# Snow redistribution in an intermediate-complexity snow hydrology modelling framework

Louis Quéno[1], Rebecca Mott[1], Paul Morin[1], Bertrand Cluzet[1], Giulia Mazzotti[1,2], and Tobias Jonas[1]

[1]WSL Institute for Snow and Avalanche Research SLF, Davos, Switzerland
[2]Univ. Grenoble Alpes, Université de Toulouse, Météo-France, CNRS, CNRM, Centre d'Études de la Neige, 38100 St. Martin d'Hères, France

**Correspondence:** Louis Quéno (queno@slf.ch)

**Abstract.** Snow hydrological regimes in mountainous catchments are strongly influenced by snowpack heterogeneity resulting from wind- and gravity-induced redistribution processes, requiring their modelling at hectometric and finer resolutions. This study presents a novel modelling approach to address this issue, aiming at an intermediate complexity solution to best represent these processes while maintaining operationally viable computational times. To this end, the physics-based snowpack model FSM2oshd was complemented by integrating the modules SnowTran-3D and SnowSlide to represent wind- and gravity-driven redistribution, respectively. This new modelling framework was further enhanced by implementing a density-dependent layering to account for erodible snow without the need to resolve microstructural properties. Seasonal simulations were performed over a 1180 $\text{km}^2$ mountain range in the Swiss Alps at 25, 50 and 100 m resolution, using appropriate downscaling and snow data assimilation techniques to provide accurate meteorological forcing. Particularly, wind fields were dynamically downscaled using WindNinja to better reflect topographically induced flow patterns. The model results were assessed using snow depths from airborne LIDAR measurements. We found a remarkable improvement in the representation of snow accumulation and erosion areas, with major contributions from saltation and suspension as well as avalanches, and modest contributions from snowdrift sublimation. The aggregated snow depth distribution curve, key to snowmelt dynamics, was significantly and consistently matching the measured distribution better than reference simulations, from the peak of winter to the end of the melt season, with improvements at all spatial resolutions. This outcome is promising for a better representation of snow hydrological processes within an operational framework.

## 1 Introduction

Snow is a crucial water resource in mountainous areas, where snowmelt represents a significant part of the runoff (e.g. Li et al., 2017). In the context of fast and marked changes of the cryosphere and water resources in the European Alps (Beniston et al., 2018), monitoring the snow cover in mountainous countries like Switzerland is necessary to assess its contribution to the streamflow in watersheds (e.g. Griessinger et al., 2019), to estimate its response to climate change in terms of runoff (e.g. Bavay et al., 2013; Hanzer et al., 2018) or rain-on-snow events (e.g. Schirmer et al., 2022), or to better anticipate future consequences on water scarcity (e.g. Brunner et al., 2019).

When estimating the state of mountain snow cover, the main challenge is to capture its seasonal evolution and strong spatial heterogeneity that occurs at different scales. Many studies have highlighted the benefits of using kilometric resolution meteorological data from numerical weather prediction models as input to snowpack models to represent most of the sources of variability (e.g. orographic precipitation) at the mountain range scale (e.g. Vionnet et al., 2016; Quéno et al., 2016; Luijting et al., 2018; He et al., 2019; Raparelli et al., 2023). At such scales, snow redistribution can usually be considered as part of sub-grid processes.

At hectometric and finer resolutions, the fine scale variability of snow distribution also has a significant impact on catchment hydrology (e.g. Luce et al., 1998). Anderton et al. (2002) showed that the decametric to hectometric variability of snow cover is critical for larger-scale snowmelt runoff simulations. Several studies emphasized that the spatial distribution of snow cover prior to the melt season is more important than spatial differences in melt behaviour for estimating cumulative snowmelt dynamics in a catchment (e.g. Anderton et al., 2004; Egli et al., 2012). Brauchli et al. (2017) identified the effects of a more heterogeneous snowpack on the melt season at the sub-basin scale, with an earlier onset of runoff and an extension of the melt season due to shallower and deeper snow-covered areas, respectively. Several redistribution processes contribute to the slope-scale variability: gravitational redistribution in steep slopes (e.g. Sommer et al., 2015; Mott et al., 2019), wind-driven snow transport (e.g. Pomeroy and Gray, 1995; Mott et al., 2018) and near-surface atmospheric effects on snowfall deposition patterns (e.g. Wang and Huang, 2017; Gerber et al., 2019). Sublimation of suspended snow can also have a significant local impact on the snowpack mass budget, although the overall contribution is usually small at the regional scale in alpine areas (Strasser et al., 2008; Bernhardt et al., 2012; Groot Zwaaftink et al., 2013; Sexstone et al., 2018). Altogether, these redistribution processes drastically alter snow distribution and their representation in snow cover models is crucial for snow hydrology beyond hectometric resolution (Clark et al., 2011).

Post-deposition snow redistribution processes, in particular wind-driven snow transport, have been studied for several decades (e.g. Dyunin and Kotlyakov, 1980; Föhn and Meister, 1983; Pomeroy and Gray, 1990), and many blowing snow models have been developed with a wide range of complexity depending on the study context and application. The complexity of blowing snow models can be broadly categorized according to the following three criteria:

- The three-dimensional turbulent diffusion equation can be resolved explicitly, as in the snowdrift module of the Alpine3D model (Lehning et al., 2008), in the Snowdrift3D model (Schneiderbauer and Prokop, 2011), in the snow2blow model (Sauter et al., 2013), in the coupled MesoNH-Crocus models (Vionnet et al., 2014) or, with a steady-state assumption, in the PBSM-3D model (Marsh et al., 2020a). To mitigate associated high computational costs, some models alternatively use a parameterization by vertical integration, as the PBSM model (Pomeroy et al., 1993), the SnowTran-3D model (Liston et al., 2007) or, more recently, the SnowPappus model (Baron et al., 2024).

- The snowpack model coupled to the snowdrift module can cover a wide range of complexity, from simple models that do not represent layer properties to detailed layered models that resolve snow microstructure. For example, studies based on SnowTran-3D (e.g. Bernhardt et al., 2009, 2010, 2012; Gascoin et al., 2013; Sexstone et al., 2018; Reynolds et al., 2020) are embedded within the SnowModel modelling framework (one-layer snowpack; Liston and Elder, 2006). Recently, a

Lagrangian multi-layer version of the latter model (SnowModel-LG) has been developed (Liston et al., 2020). Marsh et al. (2020a) associate PBSM-3D to the Snobal model (two-layer snowpack; Marks et al., 1999). Musselman et al. (2015) use PBSM within the Distributed Snow Model (three-layer snowpack). The aforementioned models solve the mass and energy budgets of the snowpack, hence providing snow layer properties such as density, temperature and liquid water content, but do not resolve the snow microstructure properties, contrary to multi-layer models like SNOWPACK (Lehning et al., 2002), used in Alpine3D (e.g. Dadic et al., 2010; Mott et al., 2010) and CRYOWRF (Sharma et al., 2023) simulations, or Crocus (Vionnet et al., 2012), e.g. used by Vionnet et al. (2014) or Baron et al. (2024). The latter snowpack models benefit from additional information on surface snow properties, which can improve the determination of snow erodibility (Guyomarc'h and Mérindol, 1998; Lehning et al., 2000), compared to formulations based on air temperature (Li and Pomeroy, 1997a) or snow density (Liston et al., 2007) used with the first category of models.

– The meteorological data used to derive wind fields driving the models can vary, from spatial and temporal interpolation of station measurements (e.g. Gascoin et al., 2013), statistical or dynamical downscaling of wind fields (e.g. Reynolds et al., 2020), deep learning methods of wind field downscaling (e.g. Le Toumelin et al., 2023), to high-resolution atmospheric models, either to produce forcing fields (e.g. Bernhardt et al., 2009; Mott et al., 2010), or a full coupling of atmosphere and surface processes (e.g. Vionnet et al., 2014; Sharma et al., 2023).

The level of complexity adopted in studies depends on the size of the simulation area (from a few square kilometers to local mountain ranges) and study duration (from individual events to full seasons). These choices are guided by the necessity to manage computational constraints and achieve a suitable model-resource equilibrium. The present study derives its objectives and constraints from the context of the Swiss Operational Snow Hydrology Service (OSHD; Mott et al., 2023), performing physics-based snow cover simulations over a large alpine domain covering the whole Switzerland, at 250 m horizontal resolution. Snow redistribution is not currently incorporated in the model. Yet, users of the OSHD simulations, such as the Swiss Avalanche Warning Service, would benefit from simulations representing slope-scale variability. We investigate here the added value of modelling snow redistribution at hectometric or smaller scales in the particular framework of intermediate-complexity snowpack modelling enabling calculations over large domains with hourly updates.

A few recent studies have explored different approaches to performing seasonal snowpack simulations, which encompass snow redistribution over large domains, all while maintaining computationally viable costs. Mower et al. (2024) parallelized SnowModel, including the SnowTran-3D module, to enable distributed snow evolution simulations at 100 m horizontal resolution over the contiguous United States. Baron et al. (2024) have chosen to use a simplified one-dimensional advection-diffusion equation in their snowdrift module SnowPappus, which is coupled to the complex multi-layer snowpack model Crocus, with a target horizontal resolution of 250 m. Vionnet et al. (2021) performed distributed snowpack simulations including parameterized gravitational redistribution (Bernhardt and Schulz, 2010) and snowdrift modelling with PBSM-3D, using a simplified three-dimensional advection-diffusion equation (Marsh et al., 2020a), with an adaptative mesh resolution (Marsh et al., 2020b). The present study introduces a different method to achieve an efficient solution: an enhanced snow cover modelling technique that comprehensively considers erodible snow layering and incorporates snow redistribution within an intermediate-complexity

framework. This combination of methods offers a novel approach, with the aim of facilitating operational applications over an entire mountain range throughout an entire winter season. After presenting modelling (Sect. 2) and evaluation methods (Sect. 3), the model will be assessed against spatially distributed snow depth measurements (Sect. 4.1), with a quantification of the impact of redistribution on the modelled snow hydrological mass budget (Sect. 4.2). Results will be discussed in Sect. 5.

## 2 Modelling methods

### 2.1 Modelling domain

The domain used for this study covers an area of 31.6 km by 37.3 km (1178.7 km$^2$) located in the eastern Swiss Alps around Davos (Fig. 1). This area covers a wide range of elevations (from 540 m.a.s.l. to 3417 m.a.s.l.), mostly in open terrain (77 %), and includes valleys and ridges of different orientations. Forests and urbanized areas were excluded from the study to focus on redistribution processes in open terrain. The prevailing wind directions in the region range from north-west to south-west.

A Digital Elevation Model (DEM) is generated on this domain at three different spatial resolutions (100 m, 50 m and 25 m) over which spatially distributed simulations are performed. The DEM is derived from the 25 m resolution DEM of the Federal Office of Topography swisstopo. The 100 m, 50 m and 25 m resolution domains contain 117868, 471472 and 1885888 grid points respectively. Figure 1 also shows the evaluation subdomains B0, D0, D1 and D2, where D1 and D2 are part of D0.

### 2.2 Snowpack modelling

FSM2 is an intermediate-complexity snowpack model (Essery, 2015; Mazzotti et al., 2020), that explicitly resolves the snowpack mass and energy balance, including fluxes between the snowpack and the atmosphere and fluxes between the snowpack and the underlying ground. However, contrary to detailed models like Crocus (Vionnet et al., 2012) or SNOWPACK (Lehning et al., 2002), it does not resolve the snow microstructural properties. This model is therefore particularly suited for advanced snow hydrological simulations, with a low computational cost (Magnusson et al., 2015). This is why a variant named FSM2oshd was developed, and is currently used within the modelling framework of the OSHD (Mott et al., 2023). The differences between FSM2 and FSM2oshd are described in detail by Mott et al. (2023). Snow-canopy interaction processes are represented in both models (Mazzotti et al., 2020), but were not considered in the present study, which focuses only on open areas. A summary of all FSM2 variants mentioned in the present study is provided in Table 1.

In order to represent erosion and accumulation due to redistribution in this intermediate-complexity framework, a few modifications were implemented. Indeed, the default layering scheme of FSM2 and FSM2oshd is a fixed stratification with predefined thicknesses (Essery, 2015), independent of snow properties (a maximum of 3 layers with top layers of 10 cm and 20 cm in FSM2oshd), which is inherently limited in representing near-surface snow evolution. Cristea et al. (2022) highlighted the significance of layering and the thickness of the upper layer in modelling accumulation and melting processes. This becomes even more important for redistribution mechanisms, as snow erodibility can change significantly throughout the snowpack stratigraphy due to varying microstructural properties. Of specific importance is the erodibility of surface snow. In cases where these

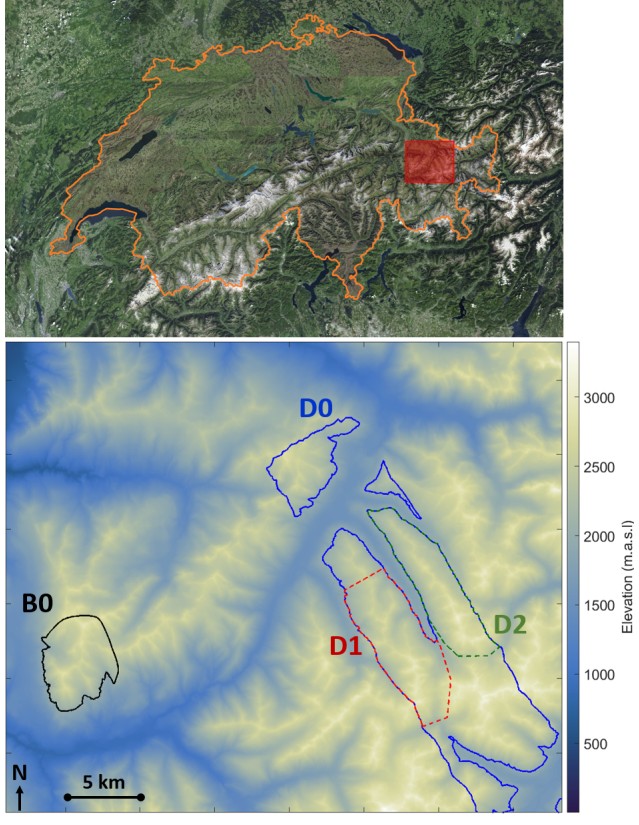

**Figure 1.** Overview of the study area: location of the domain in Switzerland (top, domain in red, Swiss borders in orange, source: swisstopo) and digital elevation model of the domain at 25 m resolution (bottom), with borders of subdomains B0, D0, D1 and D2. D1 and D2 are part of D0.

**Table 1.** Summary of the different versions of FSM2 model mentioned in the study.

| Model version | Description | References |
|---|---|---|
| FSM2 | Original snowpack model | Essery (2015), Mazzotti et al. (2020) |
| FSM2oshd | Nationwide operational implementation of FSM2 incl. data assimilation | Mott et al. (2023) |
| FSM2ref | FSM2oshd + density-dependent layering | present study |
| FSM2trans | FSM2ref + wind- and gravity-driven redistribution | present study |
| FSM2trans aval. | FSM2ref + gravity-driven redistribution | present study |
| FSM2trans wind | FSM2ref + wind-driven redistribution | present study |

microstructural attributes are not directly resolved, snow density serves as the most suitable proxy for assessing erodibility. Hence we implemented a new density-dependent layering scheme, enabling a finer layering near the surface (Fig. 2). Dynamic

layering methods based on microstructural properties are common in complex snowpack models (e.g. Vionnet et al., 2012), but the novelty of the present method is to introduce a simpler dynamic layering suitable for models like FSM2oshd.

The density-dependent layering is run at each time step and at each grid point. When snowdrift and avalanches are enabled, the relayering scheme is also run after each redistribution process. It is based on the below steps and is constrained by 3 parameters: $N_{max}$, the maximum number of layers allowed (here defined as 6), $HS_{min}$, the minimum snow layer thickness

allowed (here defined as 2 cm), and $HS_{fine}$, the maximum surface thickness where the snowpack will be finely layered (here defined as 50 cm). The layering is performed in the following sequence:

–  Every time new snow accumulates, whether it's from snowfall, snowdrift, or an avalanche, it adds a new layer to the top of the snowpack.

–  All the snow deeper than $HS_{fine}$ is moved to the basal snow layer.

–  If one of the layers is thinner than $HS_{min}$, it is merged with the adjacent layer with the closest density.

–  If there are more than $N_{max}$ layers, the two adjacent layers with the closest density are merged.

–  If the number of layers used is less than $N_{max}$, and sufficiently thick layers can be divided, a recursive process of splitting the thickest layers in two follows. This continues until the total layer count reaches $N_{max}$, adhering to the established criteria of minimum layer thickness and maintaining a finely layered surface.

The aim of this routine is to provide a fine layering near the snowpack surface to determine surface snow conditions better, while using as many density-homogeneous layers as possible to represent the snowpack historical stratigraphy without resolving the snow microstructure. Similar to the default layering routine of FSM2, mass and energy are conserved throughout the relayering steps by tracking the thickness, ice and water content and internal energy of all layers, with appropriate weighting when splitting or merging occurs.

Furthermore, wet or refrozen snow layers are identified as non-erodible (e.g. Li and Pomeroy, 1997b). To this end, a mechanism to keep track of past snow wetting events is implemented. The historical wetting variable $hist_{wet}$ of a given layer is initialized as $hist_{wet} = 0$, and set to $hist_{wet} = 1$ when this layer reaches its maximum liquid capacity leading to drainage. A weighted average is performed during relayering. A layer is considered as non-erodible by snowdrift when $hist_{wet} > 0.5$ (independently of friction velocity erodibility calculations in the snowdrift module). In the example situation illustrated in

Fig. 2, fresh snow accumulation creates a new top snow layer, snow deeper than $HS_{fine}$ is added to the basal layer and, as the fourth layer from the top gets too thin, it is merged with layer 5, of closest density. Since layer 2 has been previously wetted, the erodible snow only consists of the top low-density layer.

The snowpack simulations with the new layering scheme were compared to the operational simulations with fixed layering and showed a very similar seasonal evolution, except for slightly increased settling and melting, most likely due to the presence

of finer layers. As the FSM2oshd parameters are tuned each year and the seasonal dynamics were close, this difference was not considered significant. For clarity, the reference version of FSM2oshd including the aforementioned developments, specifically developed for the present study, is called FSM2ref hereafter (Table 1).

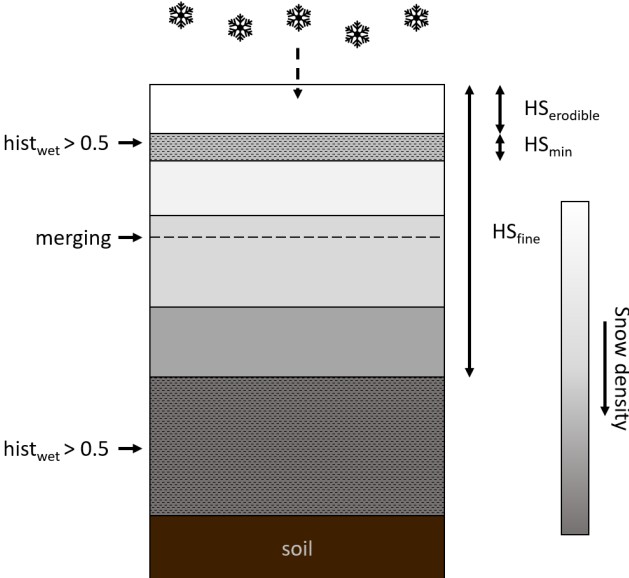

**Figure 2.** Density-dependent layering scheme implemented in FSM2ref.

## 2.3 Redistribution modelling

In the OSHD modelling framework, the spatially distributed FSM2ref model does not represent any lateral interaction between grid points in open terrain (Mott et al., 2023). Post-deposition redistribution processes were implemented within this OSHD framework. The process representation complexity needs to match the intermediate complexity of FSM2ref snowpack modelling, i.e. no snow microstructure resolving and a hourly timestep to maintain an acceptable compromise for operational simulations. With that regard, we chose to integrate and adapt SnowTran-3D (Liston and Sturm, 1998; Liston and Elder, 2006; Liston et al., 2007) for wind-driven redistribution modelling and SnowSlide (Bernhardt and Schulz, 2010) for gravity-driven redistribution modelling, in a new version of FSM2ref named FSM2trans hereafter (Table 1).

### 2.3.1 Wind-driven redistribution

SnowTran-3D is a model for wind-induced snow transport based on semi-empirical parameterizations (Liston and Sturm, 1998; Liston and Elder, 2006; Liston et al., 2007), i.e. not explicitly resolving the three-dimensional turbulent diffusion equation. Vertically integrated snow transport fluxes and sublimation rates are calculated in the saltation and the suspension layer, based on two-dimensional wind field inputs. This parameterization enables efficient computations that benefit large-scale or full-season simulations, at spatial resolutions between 1 m and 100 m, with a large span of applications from 2 m resolution simulations of local drift patterns around road beds in flat terrain (Liston et al., 2007), to 90 m resolution seasonal simulations in glaciated mountains (Gascoin et al., 2013). In particular, this scheme is well suited for intermediate complexity snowpack models, since the threshold friction velocity to initiate snowdrift is parameterized by a formulation that relates it to snow density

through exponential laws (Liston et al., 2007). Thus, this scheme was implemented within FSM2trans, with default parameter values, and without using Tabler (1975) equilibrium-snowdrift profiles, which are more suitable for higher resolutions than the scope of our study (Liston et al., 2007).

We integrated a few adaptations and improvements, described in the following. In this new implementation, snow erosion is performed layer by layer, where the eroded snow depth is derived from the mass flux using the density of each layer, rather

than assuming a constant density. However, as in the original model, redeposited snow is assigned a constant density of 300 $kg/m^3$. Indeed, in the absence of snow microstructure modelling, laws parameterizing the compaction of redeposited snow during snowdrift, as proposed by Durand et al. (2001), are irrelevant. The compaction of the top snow layer under the influence of wind is taken into account, following the original SnowTran-3D parameterization. The identification of erodible snow is assessed for each layer based on existing liquid water content, past wetting history, and the relevant threshold friction velocity.

Consequently, the SnowTran-3D conventional two-layer concept (comprising surface soft snow and underlying hard snow) is enriched to incorporate a more sophisticated scheme, ensuring a more accurate representation of the layering within the snowpack.

### 2.3.2 Gravity-driven redistribution

SnowSlide (Bernhardt and Schulz, 2010) is a model using a simple parameterization for gravitational snow transport. Avalanches

are simulated when a slope threshold and a snow holding capacity are exceeded. The snow holding capacity is defined as a threshold in snow thickness (i.e. normal to the slope), dependent on the slope. The parameterization for snow holding capacity follows that used in the implementation of SnowSlide in the Canadian Hydrological Model (CHM; Marsh et al., 2020b). The process is solved sequentially from the highest elevation pixel to the lowest one in the domain. Snow exceeding the holding capacity is transported laterally to lower neighbouring pixels, proportionally to their elevation difference. The physics of

avalanches, from triggering to dynamics, is not explicitly solved. This scheme was implemented within FSM2trans with a few improvements to mitigate these limitations, in the form of simple hysteretic features, described in the following. In this new implementation, the slope threshold of 25 degrees is only used for avalanche triggering. However, no such threshold is applied to pixels that receive the snow released by avalanches, to enable a larger deposition area. Furthermore, the snow holding capacity is decreased by 30% for the timesteps a pixel receives an avalanche, to mimic avalanche dynamics.

In its original version, SnowSlide updates the DEM with the newly calculated snow depth at each time step, which allows to update the slope and the order of the pixel calculations sorted by decreasing elevations. This version has been tested with no significant visible differences in avalanche deposition areas. However, the calculation of the new sorted elevation list at each time step has a high computational cost. Consequently, we decided to discard this step with a view to intermediate complexity modelling applicable to operations. The implementation of the hysteretic features showed a more significant impact on the

avalanche extent.

### 2.3.3 Integration of the redistribution submodels in FSM2trans

The adapted SnowTran-3D and SnowSlide submodels are integrated as subroutines within FSM2trans. After solving the one-dimensional snowpack processes (heat conduction, melting, sublimation, water percolation, compaction, fresh snow addition), the layering subroutine is called, followed by wind-driven redistribution, relayering, gravity-driven redistribution and a final relayering. SnowSlide follows SnowTran-3D, as snowdrift occurs at the exact timestep (under the given snow and meteorological conditions), while avalanches may actually occur with a certain delay. It enables to take into account the potential avalanche triggering on slopes loaded by snowdrift, and to avoid the immediate gravitational transport of fresh snow to lower elevations, where snowdrift is less likely to happen.

Each submodel can be activated independently via a namelist. The FSM2trans model version with gravity-driven redistribution only is called FSM2trans aval. while the version with wind-induced redistribution only is called FSM2trans wind (Table 1).

## 2.4 Meteorological input

FSM2ref and FSM2trans require several meteorological inputs, for the simulated grid points, at an hourly time step: near-surface air temperature, relative humidity, and wind speed (the three of them at a defined height above the ground), longwave irradiance, direct and diffuse shortwave irradiance, air pressure, rainfall and snowfall. The snowdrift module of FSM2trans also requires the wind direction. Similar to the operational version FSM2oshd (Mott et al., 2023), these fields are primarily derived from the hourly analysis fields of the regional weather forecast model COSMO with a spatial resolution of 1 km. To enhance the effect of fine-scale topographical influences, this information is subsequently downscaled to 100 m, 50 m and 25 m spatial resolutions. In particular, near-surface air temperature, relative humidity and air pressure are downscaled by linear interpolation with lapse rates. Direct and diffuse shortwave irradiances are dynamically downscaled following the approach of Jonas et al. (2020). Longwave irradiance downscaling follows Helbig and Löwe (2014). Snow depth measurements at stations are assimilated to improve the solid precipitation estimate through a data assimilation scheme using optimal interpolation (Magnusson et al., 2014). Total precipitation is then linearly interpolated to the finer grid where the phase split between rain and snow is made according to the downscaled near-surface air temperature field, following the same formulation as FSM2oshd using a sigmoid function centered on a 10 m air temperature of 1.04 °C.

Snowdrift is strongly determined by local topographic effects on wind fields (e.g. Mott et al., 2018). Consequently, the downscaling of wind patterns must encompass the interplay between terrain and airflow dynamics, such as local acceleration and deceleration near ridges, or channeling in valleys and gullies. For this purpose, we used the mass-conserving dynamical downscaling model WindNinja (Forthofer et al., 2014; Wagenbrenner et al., 2016), version 3.7.0, which was forced by COSMO 1 km resolution wind fields. WindNinja was run separately at 100 m, 50 m and 25 m spatial resolutions to produce downscaled wind fields for the snowpack simulations, providing horizontal wind speed and direction. Similar to Vionnet et al. (2021), the mass- and momentum-conservation option (Wagenbrenner et al., 2019) was not free of model instabilities on the study area with complex topography and was therefore not retained.

### 2.5 Simulation setup

Spatially distributed seasonal snowpack simulations were performed with FSM2ref and FSM2trans at hourly timestep over the entire study area, from 1 September 2016 to 30 June 2017 and from 1 September 2019 to 30 June 2020, at resolutions of 25, 50 and 100 m. To further assess the distinct effects of avalanche and snowdrift modelling, complementary simulations were performed in a similar setup with FSM2trans aval. and FSM2trans wind.

### 3 Evaluation data and methods

Four distinct datasets of distributed snow depth measurements were used to evaluate the snowpack simulations. These measurements were acquired by airborne LIDAR technology. During the 2016-2017 hydrological year, three aerial surveys covered a region centered on the Dischma valley near Davos, with a spatial resolution of 1 metre. To exclude forests and urbanized areas, we filtered out elevations under 2000 m.a.s.l. (corresponding to the treeline), since the forest snow model instance of FSM2oshd was not used in the present study. The resulting subdomain is named D0 (Fig. 1). The airborne surveys took place

on three dates: 20 March 2017, which marked the transition from winter to the melt season; 31 March 2017; and 17 May 2017, the latter two dates covering the melting period. These datasets were validated against more than 11 thousand manual measurements, resulting in a bias of - 4 to 0 cm and a root-mean-square deviation (RMSD) of 4 to 8 cm (Mazzotti et al., 2019). During the 2019-2020 hydrological year, one airborne survey covered an area east of Lenzerheide, with a spatial resolution of 1 metre. Similarly, elevations under 2000 m.a.s.l. were filtered out, resulting in subdomain B0 (Fig. 1). The flight was conducted on 17

March 2020 (at the transition between winter and melt season). This dataset was validated against 79 manual measurements, resulting in a bias of - 2 cm and a RMSD of 15 cm.

The datasets were post-processed to mask out lakes and a few obvious outliers (e.g. due to buildings or in very steep terrain). The resulting snow depth maps were then aggregated to resolutions of 25 m, 50 m and 100 m. As the outliers were mostly below 2000 m.a.s.l. and very isolated above 2000 m.a.s.l., the masked data at high resolution were simply excluded from the

mean over aggregated pixels. In a second step, glacier masks were applied to the averaged snow depth maps.

Simulation results were extracted on the whole domain and on the masked LIDAR subdomains B0, D0, D1 and D2 (Fig. 1). Modelled snow depth maps were compared to LIDAR snow depth maps upscaled to the model grid at 25 m, 50 m and 100 m resolution, respectively. Beyond visual comparison, results were aggregated by elevation bands and aspect to identify distribution patterns better. In addition, results were aggregated by Topographic Position Index (TPI), a parameter that characterizes

the relative height of a point in relation to its local surroundings. It was calculated using a 25 m resolution DEM with a 2 km radius neighbourhood. This terrain descriptor is particularly suited for a focused analysis of areas most susceptible to wind exposure: it is often used in wind downscaling methods, e.g. by Winstral et al. (2017) and Dujardin and Lehning (2022). For this purpose, we defined a "ridges" category for pixels with a TPI exceeding 200 m. This classification facilitates a specific analysis of these areas of interest.

To complement the visual comparison of simulated and measured snow depth maps, we introduced a quantitative approach based on the Structural Similarity Index (SSIM; Wang et al., 2004). This metric was originally developed for image quality

assessment to quantify the similarity between a distorted (e.g. compressed) image and a reference image. It is a combination of luminance, contrast and structure comparison. Pixels are compared with their neighbourhood using a Gaussian weighting function. We apply it to snow depth maps by considering them as grayscale images, where the snow depth is the intensity on a scale from 0 to 5 m, using a Gaussian radius of 150 m. The mean SSIM (MSSIM) indicates the overall similarity of the two snow depth maps. A random snow depth distribution in the 0 to 5 m range gives $MSSIM = 0$. Values closer to 1 indicate better similarity, and $MSSIM = 1$ if and only if the snow depth maps are identical. The Structural Similarity Index can be computed using the ssim function from the Matlab Image Processing Toolbox (https://www.mathworks.com/help/images/ref/ssim.html, last access: 8 May 2024), or using the scikit-image library in Python (https://scikit-image.org/docs/stable/auto_examples/transform/plot_ssim.html, last access: 8 May 2024).

## 4 Results

### 4.1 Comparison of simulated snow depth to LIDAR data

Initially, we present snow depth maps derived from FSM2trans simulations, offering a comparative analysis against both LIDAR datasets and reference FSM2ref simulations that do not incorporate redistribution effects. This analysis illustrates the capabilities of FSM2trans to represent specific redistribution patterns, at different stages of the snow season, from the peak of accumulation to the melt season. These maps are presented for subdomains to assess local patterns with more clarity. Figure 3 shows the snow depth map for subdomain B0 on 17 March 2020, while Fig. 4 shows subdomain D2 on 31 March 2017 and Fig. 5 subdomain D1 on 17 May 2017. We highlighted specific erosion and accumulation patterns with red arrows on the maps.

A necessary element for the interpretation of these maps is the elevation profile of snow depth at peak of winter for subdomain D0 (Fig. A1 in the Supplementary Material). It shows a marked underestimation of snow depth by FSM2ref compared to the LIDAR at elevations above 2600 m.a.s.l., despite the absence of redistribution and snowdrift sublimation that would further reduce mean snow depths at the highest elevations. This suggests a significant underestimation of solid precipitation at high elevations in the forcing of FSM2ref and FSM2trans, which is consistent with the findings of Mott et al. (2023).

Although FSM2trans produces too little snow at the highest elevations, the smaller scale snow distribution patterns match observations well. In Fig. 3, arrow 1 indicates three avalanches following steep and narrow gullies. These avalanches have been well simulated by FSM2trans, and in particular, their deposition areas show a good match with observations. Arrow 2 shows another avalanche that was successfully simulated. However, the deposition area is underestimated by FSM2trans. This underestimation of avalanche area can be noted for several other cases. The location of simulated high snow accumulation behind ridges exhibits a remarkable overall agreement with the observed data. This accumulation results from a combination of snowdrift deposition and subsequent avalanche initiation within these strong accumulation zones in steep terrain. Such cases are highlighted for example by arrows 3 and 4 (Fig. 3), arrows 6 and 7 (Fig. 4), and arrows 8 and 9 (Fig. 5). The particular example of arrow 6 indicates an accurate location of a large accumulated snow mass transported by wind to the northeastern lee side of the ridge and carried by avalanches to lower elevations, extending further than the immediate foot of the steep slopes. However, the extension of the deposition areas in the simulations sometimes remains too limited, with an uncertainty to

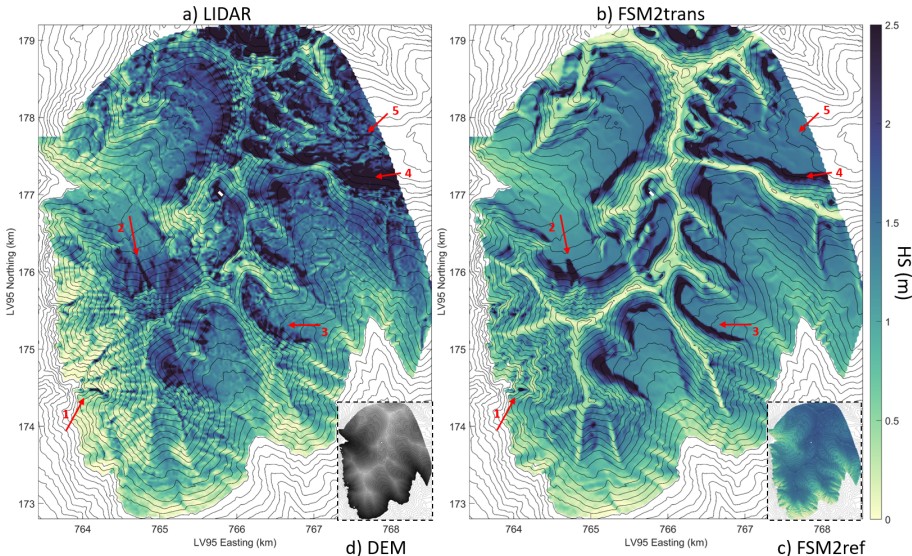

**Figure 3.** Map of snow depth on 17 March 2020 for subdomain B0: a) as measured by LIDAR and aggregated to 25 m resolution, b) as simulated at 25 m resolution by FSM2trans, c) as simulated at 25 m resolution by FSM2ref. d) Indicative Digital Elevation Model (DEM) of subdomain B0 with higher elevations in lighter gray.

attribute this shortcoming to the snowdrift or avalanche model (e.g. arrows 4 and 7). These results are consistent throughout the season, with deposition patterns being particularly visible in spring (Fig. 5). High-elevation ridges show strong erosion patterns in the simulations, which are consistently overestimated compared to the observations. Strong variability in intermediate slopes is sometimes underestimated by the model (e.g. arrow 5 in Fig. 3). The high local variability is still partially represented on a high elevation pass particularly exposed to wind (arrow 10 in Fig. 5).

The added value of the FSM2trans representation of redistribution processes is particularly noteworthy when compared to the maps resulting from FSM2ref simulations (thumbnails in Fig. 3, 4 and 5). The snow depth simulated by FSM2ref above 2000 m.a.s.l. is notably homogeneous at the end of winter (Fig. 3), with variability introduced mainly throughout the melt season due to differences in melt energy between slopes (Fig. 5). These maps confirm that simulations that do not include redistribution processes cannot represent a significant part of the snowpack spatial variability at 25 m resolution, even if Mott et al. (2023) showed that such simulations (performed with the FSM2oshd variant) captured the average state of the snowpack well when compared to station measurements at all elevation bands, except the highest elevations where snow measurements and data assimilation were lacking.

Figure 6 shows MSSIM values of snow depth maps simulated by FSM2trans, FSM2trans aval., FSM2trans wind and FSM2ref, compared to the four LIDAR datasets. Simulations at resolutions of 25, 50 and 100 m respectively are represented with increasing transparency (darkest shading for 25 m). At 25 m resolution, the highest similarity is obtained for FSM2trans simulations (0.39 to 0.45) and the lowest similarity for FSM2ref simulations (0.14 to 0.18). FSM2trans always has a higher MSSIM than FSM2trans aval. (0.35 to 0.38) and FSM2trans wind (0.19 to 0.24), highlighting the importance to model the

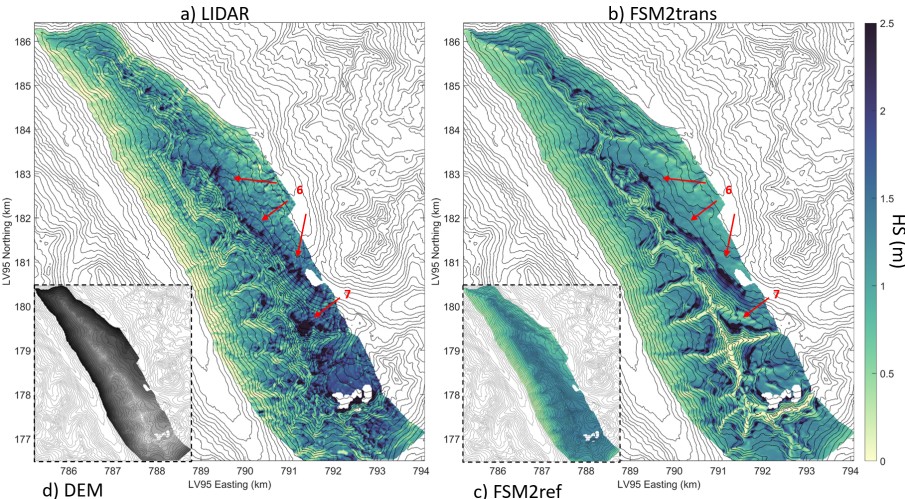

**Figure 4.** Map of snow depth on 31 March 2017 for subdomain D2: a) as measured by LIDAR and aggregated to 25 m resolution, b) as simulated at 25 m resolution by FSM2trans, c) as simulated at 25 m resolution by FSM2ref. d) Indicative DEM of subdomain D2 with higher elevations in lighter gray.

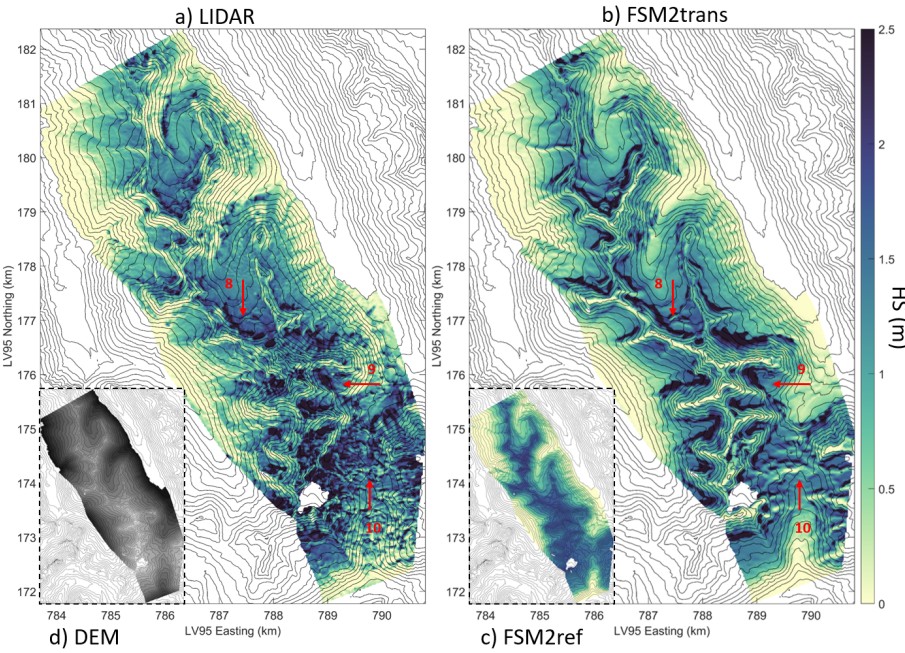

**Figure 5.** Map of snow depth on 17 May 2017 for subdomain D1: a) as measured by LIDAR and aggregated to 25 m resolution, b) as simulated at 25 m resolution by FSM2trans, c) as simulated at 25 m resolution by FSM2ref. d) Indicative DEM of subdomain D1 with higher elevations in lighter gray.

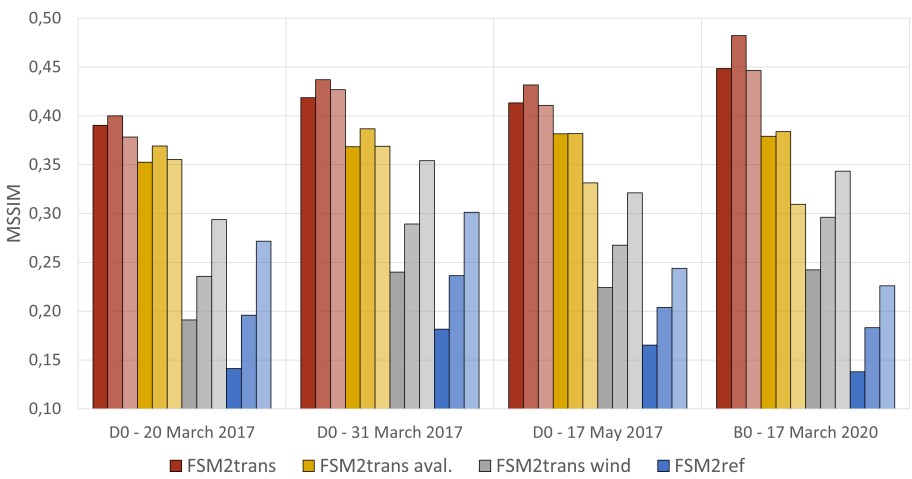

**Figure 6.** MSSIM for simulation results from FSM2trans (red), FSM2trans aval. (yellow), FSM2trans wind (gray) and FSM2ref (blue) against the four LIDAR datasets. Lighter colour shades indicate lower resolutions (25 m for darkest shading, 50 m for intermediate shading, 100 m for lightest shading).

interplay of avalanches and snowdrift. The MSSIM is higher when only avalanches are represented than when only snowdrift is represented: in a domain with a lot of steep terrain, simulating avalanches is easier because they are confined to steep slopes,

whereas snowdrift is more widespread, hence a less obvious spatial structure. At coarser resolutions, the MSSIM increases for FSM2ref simulations: the reference LIDAR map is smoother, so the similarity to simulations with less spatial variability increases. The MSSIM of FSM2ref remains clearly inferior to all other simulations representing one or both redistribution processes, at all resolutions, proving there are still clear benefits of modelling redistribution even at 100 m resolution. Following the same logic as FSM2ref, the MSSIM of FSM2trans wind increases with coarser resolutions. However, the MSSIM of

FSM2trans aval. decreases from 50 m to 100 m resolution, reaching values similar to those of FSM2trans wind: with smoother terrain, some avalanche couloirs are not represented anymore and slopes are lower, hence less avalanche triggering.

Figure 7 shows the snow depth distribution on subdomains B0 and D0 for the four LIDAR datasets and the snow depth distribution simulated by FSM2ref and FSM2trans. These plots quantitatively confirm the visual assessment of redistribution patterns. In FSM2ref (blue curve), a frequency peak is observed in all plots throughout the season, even if the distribution gets

flatter with differential melting during spring. In particular, compared to LIDAR observations (black curve), low snow depths are underrepresented, and high snow depths (typically more than 2 m) are absent. FSM2trans (red curve) clearly improves the snow depth distribution with a flatter curve matching the LIDAR curve better. Low and high snow depths are better represented, even though the spread remains lower than in observations. Discrepancies with observations are further influenced by uncertainties in precipitation input and modelling of compaction and melting processes.

In order to focus on the most wind-exposed areas, Fig. 8 represents the same frequency plots restricted to the areas where TPI > 200 m, i.e. ridges and their surroundings. The match of FSM2trans with the LIDAR is even better than when all TPIs are

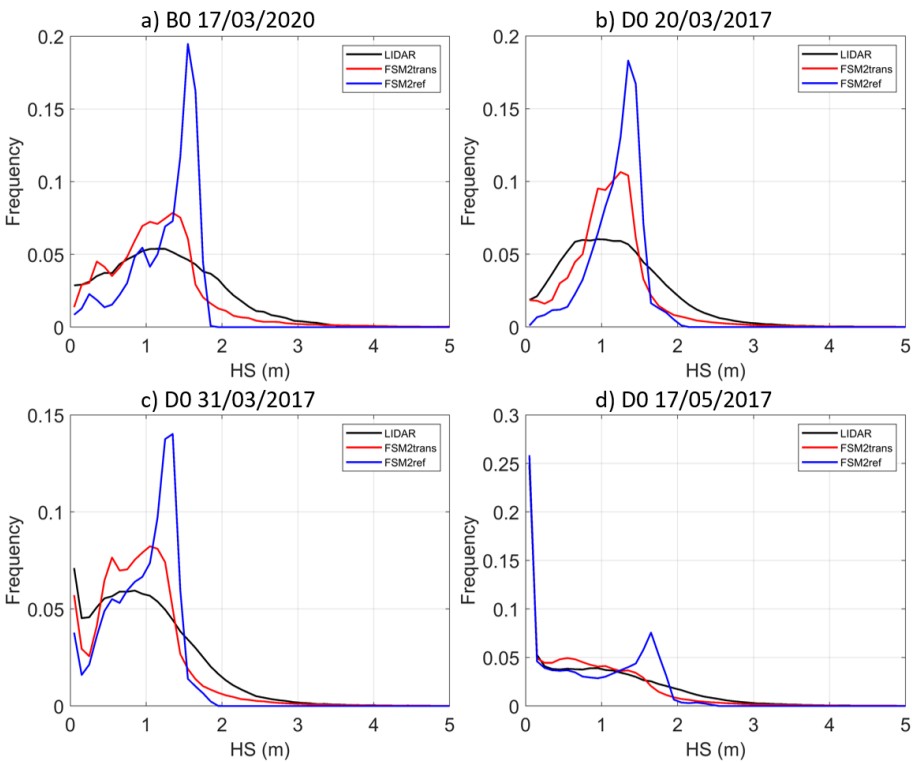

**Figure 7.** Snow depth frequency plot by intervals of 10 cm, at 25 m resolution, for LIDAR measurements (black), FSM2trans simulation (red) and reference FSM2ref simulation without redistribution (blue), for subdomain: a) B0 on 17 March 2020, b) D0 on 20 March 2017, c) D0 on 31 March 2017, d) D0 on 17 May 2017.

considered, with a clear improvement compared to FSM2ref. In the four cases, shallow snowpacks are slightly overrepresented by FSM2trans, which confirms the visual observation of excessive ridge erosion.

FSM2trans simulations at 50 m and 100 m grid resolution have also been assessed. Corresponding maps and plots are
included in the Supplementary Material (Appendices B and C). Despite the lower spatial resolution, redistribution patterns remain consistent between resolutions and compared to the LIDAR data. Fine-scale redistribution patterns are logically less present due to the absence of fine-scale terrain features, like narrow gullies channelling avalanches. However, even at 100m resolution, the snow depth frequency curve shows a similar improvement to the higher resolution simulations compared to FSM2ref (Fig. 9), which confirms the outcomes of the structural similarity analysis.

**4.2   Cumulated effect of snow redistribution**

In this analysis, we examine the cumulative effects of each redistribution process over the course of a winter season, aiming to identify their individual contributions to the snowpack dynamics while also exploring their interrelationships.

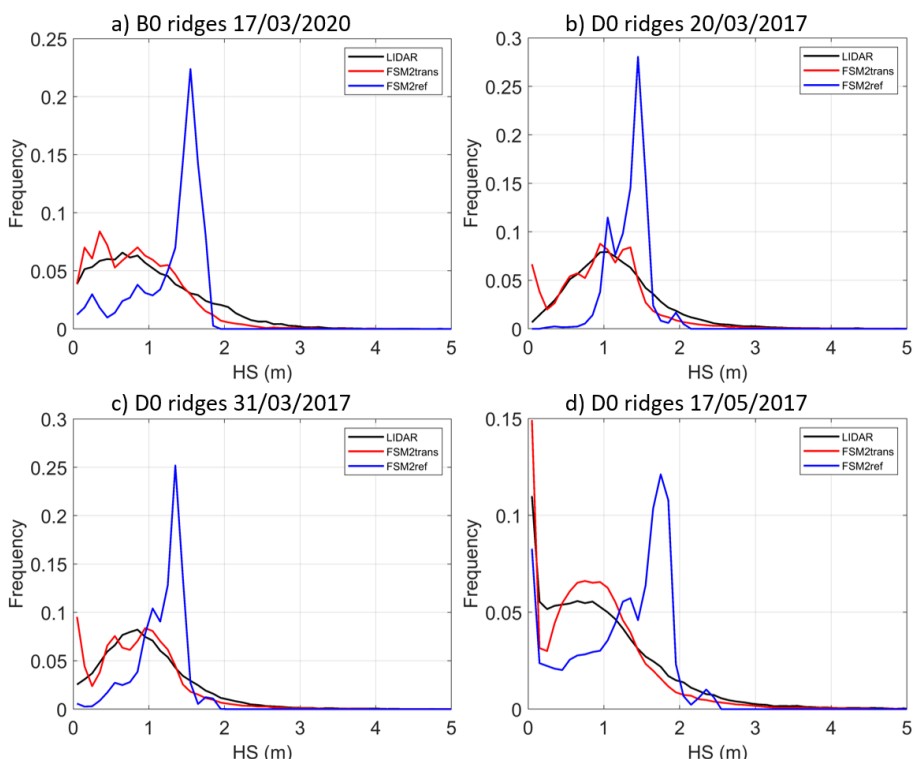

**Figure 8.** Snow depth frequency plot by intervals of 10 cm, at 25 m resolution, for LIDAR measurements (black), FSM2trans simulation (red) and reference FSM2ref simulation without redistribution (blue), for areas where TPI > 200 m, for subdomain: a) B0 on 17 March 2020, b) D0 on 20 March 2017, c) D0 on 31 March 2017, d) D0 on 17 May 2017.

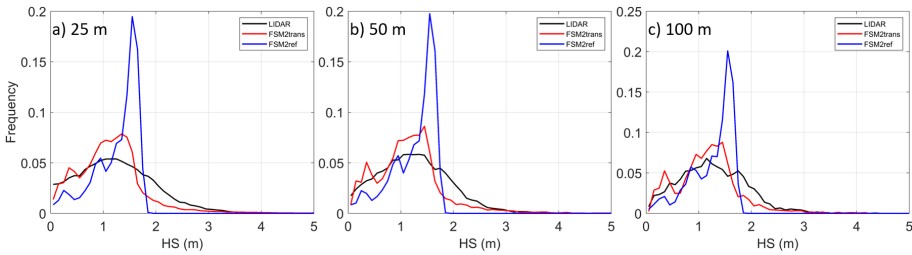

**Figure 9.** Snow depth frequency plot by intervals of 10 cm, for subdomain B0 on 17 March 2020, for LIDAR measurements (black), FSM2trans simulation (red) and reference FSM2ref simulation without redistribution (blue) at: a) 25 m resolution, b) 50 m resolution, c) 100 m resolution.

Figure 10 represents the net seasonal effect of four distinct processes on SWE within the designated subdomain D1, spanning the period from 1 September 2016 to 30 June 2017. These processes encompass: a) saltation and suspension, b) avalanches, c) snowdrift sublimation, and d) surface sublimation due to turbulent fluxes. Avalanches locally contribute very strongly to SWE change (up to 1000 mm gain or loss). The extension of avalanche effects is yet spatially restricted to steep slopes and their

immediate surroundings. The cumulative impact resulting from snowdrift-induced saltation and suspension can lead to gains or losses exceeding 500 mm. Notably, these substantial changes in SWE are most pronounced upon or in close proximity to ridgelines. The extension of saltation and suspension effects is yet more widespread than avalanches, with changes in SWE also occurring on intermediate slopes, but very limited down the valleys (typically less than 10 mm). The location of snow accumulation due to saltation and suspension is consistent with the prevailing north-west to south-west wind directions, with more frequent net SWE gains on the eastern side of the ridges. Strong southwesterly foehn wind events can also explain the erosion on the southern slopes. SWE losses due to snowdrift sublimation cover a similar spatial extent, with a lower quantitative impact (up to 100 mm on the ridges). When compared to surface sublimation due to turbulent fluxes, snowdrift sublimation locally reaches higher extreme values, while surface sublimation is more intense in valleys and low elevations. Surface deposition dominates in high elevations (apart from ridges), which tends to partly compensate for snowdrift sublimation losses.

Figure 11 illustrates the proportional influence of each individual process in relation to the total snowfall occurring between 1 September 2016 and 30 June 2017. Values are aggregated over the whole domain, encompassing 8 aspects and 12 elevation bands each spanning a 100 m range above 2000 m.a.s.l. In the most wind-exposed regions, specifically the northwestern to southwestern aspects at higher elevations, a combination of saltation and suspension processes contribute on average to a loss of approximately - 50 % of the total snowfall. On wind-sheltered aspects, accumulations due to saltation and suspension represent on average up to approximately 25 % of the total snowfall. The areas with the strongest avalanche erosion are located preferentially in areas where snowdrift accumulations are prominent and reach on average approximately - 50 % of the total snowfall there. Avalanche deposits are more widespread across elevations, so their average represents approximately 5 % of the total snowfall. Snowdrift sublimation shows an increasing trend with elevation, independently of the aspect, and reaches on average up to - 5 % of the total snowfall at the highest elevations. Ridges locally show extreme snowdrift sublimation values of - 20% of the total snowfall. The areas of strongest snowdrift sublimation correspond to areas with the lowest intensity of surface sublimation or deposition due to turbulent fluxes (deposition on high elevation northern slopes is up to 2 % of the total snowfall), while surface sublimation can reach up to - 8 % of the total snowfall at lower elevations. When considering all elevations of the whole domain, snowdrift sublimation represents a - 1.0 % loss of the total snowfall in 2016-2017 (- 1.4 % in 2019-2020), that is a smaller contribution than surface sublimation loss (- 4.3 % in 2016-2017 and - 3.5 % in 2019-2020). These contributions become similar when considering only elevations above 2000 m.a.s.l.: - 1.8 % in 2016-2017 and - 2.3 % in 2019-2020 for snowdrift sublimation; - 2.9 % in 2016-2017 and - 2.1 % in 2019-2020 for surface sublimation. Snowdrift sublimation significantly dominates at elevations above 3000 m.a.s.l.: - 4.4 % in 2016-2017 and - 3.9 % in 2019-2020, against - 0.2 % in 2016-2017 and - 0.5 % in 2019-2020 for surface sublimation.

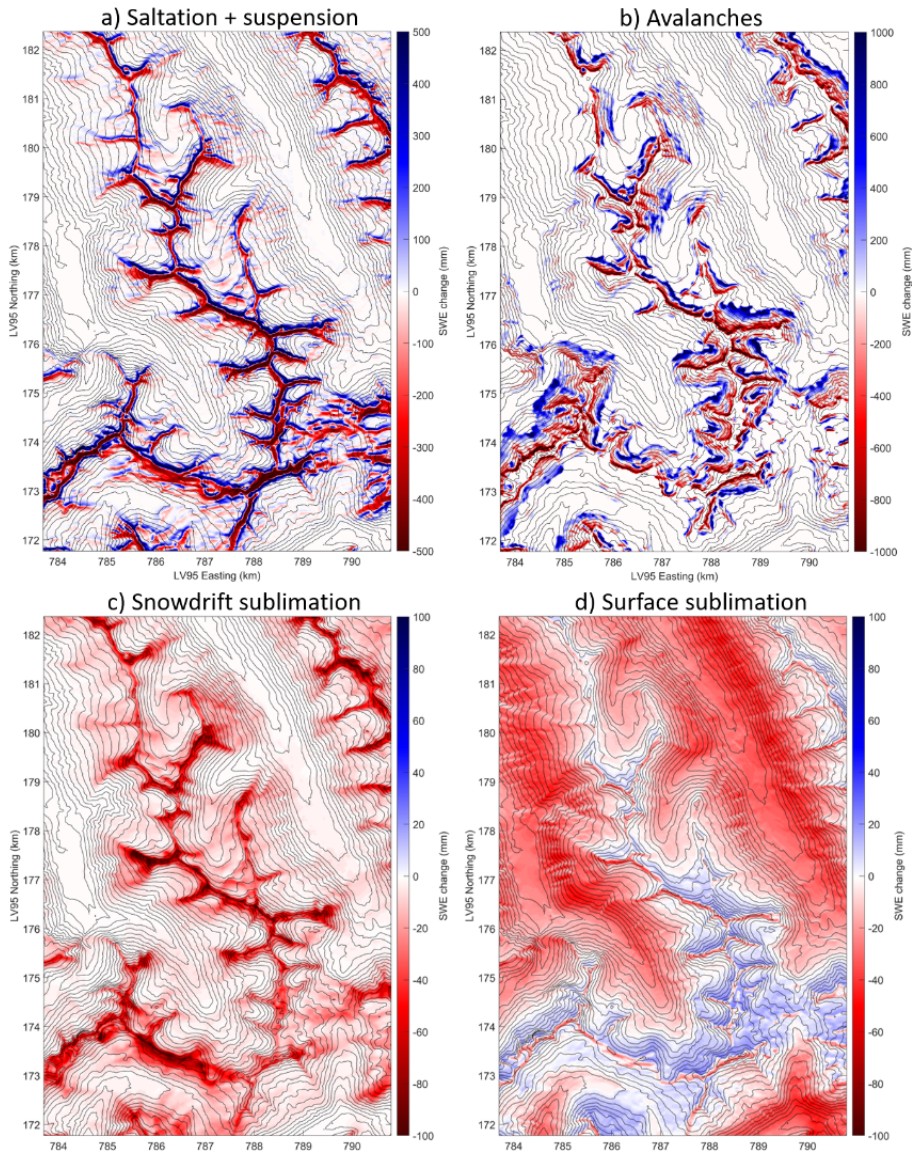

**Figure 10.** Map of net SWE change, cumulated from 1 September 2016 to 30 June 2017, for subdomain D1, as simulated by FSM2trans, due to: a) saltation and suspension, b) avalanches, c) snowdrift sublimation, d) surface sublimation. Note that colour axes differ between panels.

# 5   Discussion

## 5.1   Added value of snow redistribution modelling

We have implemented a modelling approach to capture wind- and gravity-driven snow transport within a spatially distributed
snow cover framework, used for operational snow hydrological modelling. Given that the primary objective of this implemen-

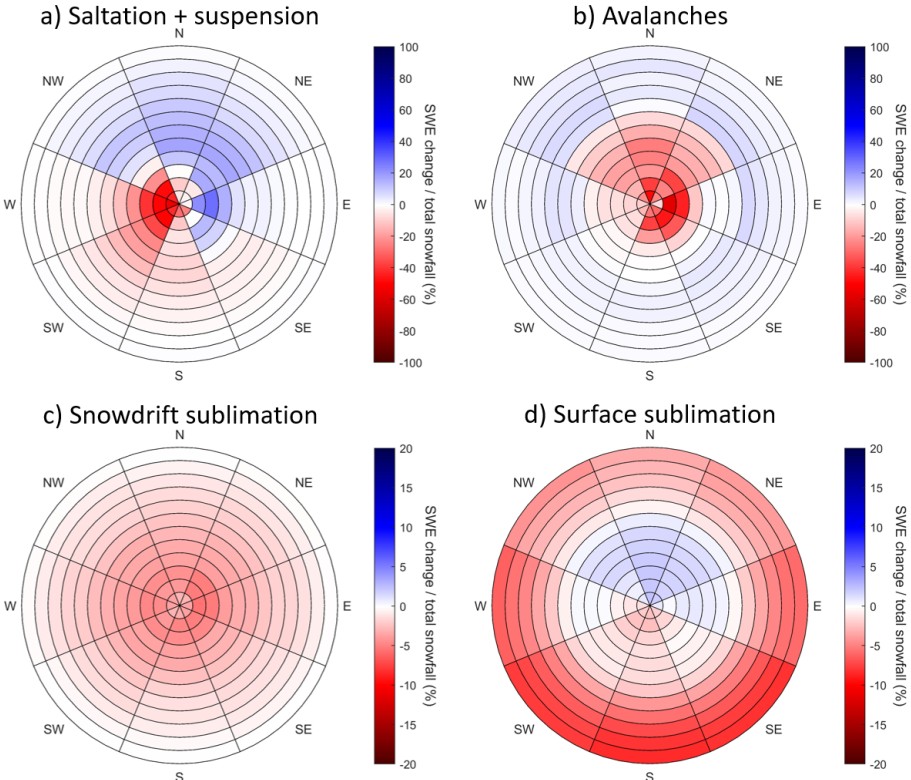

**Figure 11.** Net SWE change relative to total snowfall, cumulated from 1 September 2016 to 30 June 2017, as simulated by FSM2trans, aggregated for the whole domain by aspect and 100 m elevation band above 2000 m.a.s.l., due to: a) saltation and suspension, b) avalanches, c) snowdrift sublimation, d) surface sublimation. Note that colour axes differ between panels.

tation was to improve snow distribution patterns, we conducted a comprehensive evaluation by comparing it with LIDAR snow depth maps. The outcomes of this evaluation reveal notable enhancement in the ability of FSM2trans in accurately depicting the formation of strong snow accumulations, positioned on the correct slopes of ridges. In contrast, FSM2ref represents a very homogeneous snowpack across elevations exceeding 2000 m.a.s.l. with no aspect differences apart from differentiated melting (Fig. 3, 4 and 5). Certain accumulation and erosion patterns can be clearly attributed to either snowdrift or avalanches. Examples include the marked erosion of ridges due to strong winds and the concentration of snow slides within steep couloirs. However, it is important to acknowledge the interplay between snowdrift and avalanches: for example, strong accumulations due to wind-induced transport can create an overload triggering or enhancing an avalanche, as can be seen in many upper slopes of the maps. This interdependence justifies the need to combine snowdrift and avalanche modelling, and a global assessment of all redistribution processes when compared to spatially distributed snow depth measurements. The structural similarity analysis of snow depth maps confirmed and quantified increased spatial similarity to LIDAR data for both avalanche and snowdrift modelling, with the best improvement when both processes are combined.

Snow depth frequency plots highlight the strong added value of FSM2trans, becoming particularly evident from the onset of the melt season onward. This added value is most prominently observed in the upper slopes. The better match of FSM2trans

snow depth distribution curve with the LIDAR data at peak of winter (Fig. 7a and 7b) is a significant progress towards a better determination of catchment snow hydrological regimes, as inferred by e.g. Anderton et al. (2004), Egli et al. (2012) and Brauchli et al. (2017). It is confirmed by the persistence of this improvement in late melt season curves (Fig. 7d), when the spatial fluctuations in melt energy become a critical element for the progressive snow cover depletion (DeBeer and Pomeroy, 2017). The spatial heterogeneity of melt energy is represented in our model through a fine downscaling of the meteorologi-

cal input (in particular incoming radiation) and a representation of the effects of terrain features (like slope, aspect, surface roughness) on the snowpack energy budget. The addition of snow redistribution allows for the presence of locally strong snow accumulations and eroded areas which further influence the timing of snowmelt.

The most realistic redistribution patterns were obtained with 25 m resolution simulations. It is important to note, however, that a significant positive influence on snow distribution remained even at 50 m and 100 m resolutions, as quantified by

the structural similarity analysis. This result is encouraging for the potential application of such modelling approaches in an operational framework, as it allows the inclusion of redistribution effects in a computationally efficient manner, enabling large domain or ensemble simulations that are often required in the context of data assimilation. For studies focusing on specific limited areas of interest, conducting simulations at a 25 m resolution could still be viably executed by employing nested runs within larger-scale simulations. In that regard, the recent development of an unstructured triangular mesh (Marsh et al., 2018)

within a snowdrift-resolving snowpack model (Marsh et al., 2020a, b; Vionnet et al., 2021) offers a promising alternative solution.

Ultimately, the FSM2trans modelling framework retains its efficiency in operational contexts, requiring only a marginal increase in computational time compared to FSM2ref. For reference, performing a full seasonal simulation (i.e. ten months from 1 September to 30 June) on the present study domain (Fig. 1), at hourly time steps, on a personal computer, without code

parallelization, yields the following approximate time frames:

- 100 m resolution: 1 h 40 min for FSM2trans (+ 10% compared to FSM2ref)

- 50 m resolution: 6 h 20 min for FSM2trans (+ 20% compared to FSM2ref)

- 25 m resolution: 25 h 30 min for FSM2trans (+ 30% compared to FSM2ref)

The relatively modest increase in computational time can be explained by the fact that the modelling chain is highly com-

putationally bound by input and output processing steps. The computation times of FSM2trans are less than the computation times required to generate downscaled wind fields using WindNinja, parallelized over 8 cores (a total of 91 h per year including all three resolutions).

## 5.2 Limitations

The model assessment has pointed to a number of limitations. These limitations mostly stem from the necessary trade-offs and

adjustments inherent in accommodating an intermediate complexity framework for snow cover modelling.

First, the redistribution patterns show some shortcomings that persist across all dataset comparisons. Both maps and distribution curves consistently reveal excessive snow erosion along ridges. A first explanation is the effect of grid resolution on snowdrift modelling. Small-scale topographic features locally enable the retention of snow in saltation on windward slopes, which cannot be represented at resolutions of 25 m or coarser (Mott and Lehning, 2010). Moreover, in the absence of subgrid parameterization, the extent of strong ridgeline erosion is likely overestimated since the modelled ridgeline pixel is 25, 50 or 100 m wide. Previous studies have also demonstrated that blowing snow models reliant on two-dimensional wind inputs, such as SnowTran-3D, are notably influenced by the specifics of the input wind fields (e.g. Musselman et al., 2015). Consequently, the observed excessive snow erosion on the ridges could be attributed to WindNinja's simulation of strong wind speeds. A preliminary evaluation of WindNinja wind speeds against measurements from 13 automated weather stations positioned across the study domain, mostly at high elevations, for the month of March 2017, showed a positive bias of 1.1 m/s. Given WindNinja's established proficiency to accurately capture ridge accelerations (Forthofer et al., 2014), it is plausible that this overestimation is even more pronounced at ridge tops. Moreover, the absence of momentum conservation in the version of WindNinja used in this study (Sect. 2.4) prohibits the modelling of lee side recirculation (Wagenbrenner et al., 2016), with consequences on the blowing snow redistribution patterns. The observed deficiency in the extent of snowdrift deposition behind ridges could be attributed to the omission of these crucial terrain-induced influences on wind fields. To tackle that issue, Vionnet et al. (2021) used wind libraries in conjunction with WindNinja (Marsh et al., 2023), introducing a modification to mitigate wind speeds on lee sides. Upcoming research with FSM2trans will address the sensitivity of redistribution modelling to wind downscaling techniques.

Secondly, errors in the meteorological input propagate to the ultimate snow distribution. For example, the lack of snow on ridges may also partly arise from the lack of simulated precipitation at high elevations. This can be attributed to the fact that the precipitation input derived by optimal interpolation has been effective in mitigating the COSMO precipitation forecast bias; however, it still retains a negative bias above about 2500 m.a.s.l. (Mott et al., 2023) where the assimilated station data become sparse. Moreover, near-surface winter precipitation processes, like preferential deposition of snowfall (Lehning et al., 2008), were not accounted for in the present study, despite their significant impact near the ridges (Gerber et al., 2019) and subsequent effect on post-deposition processes. All in all, the use of a high-resolution atmospheric model may be necessary to further improve the precipitation input at such resolutions in complex terrain. The atmospheric downscaling model HICAR (Reynolds et al., 2023), recently developed with a focus on computational efficiency in complex terrain, is a promising solution to provide precipitation and wind field inputs to FSM2trans that better account for complex atmosphere-topography interactions.

Finally, the coordinated use of modified versions of SnowTran-3D and SnowSlide within our modelling framework has shown satisfactory outcomes with regard to the initial objective of improving the heterogeneity of the snow cover distribution, outweighing the shortcomings resulting from the simplified parameterizations in each model. If spatially distributed snow depth measurements (e.g. acquired by airborne LIDAR surveys) enable to evaluate the position and extent of erosion and deposition zones, more validation data is needed to assess the amount of snow lost by sublimation in the suspension layer. Our results (an average loss of approximately 1 %, reaching on average 4 % at high elevations) are consistent with previous studies based on SnowTran-3D in alpine terrain, e.g. 4.1 % for Strasser et al. (2008), 1.6 % for Bernhardt et al. (2012) and 3.4 % for

Sexstone et al. (2018). However, the parameterization of SnowTran-3D does not model the atmospheric feedbacks due to the latent heat exchange of snowdrift sublimation. Groot Zwaaftink et al. (2013) showed the feedback processes largely reduce the snowdrift sublimation down to 0.1 %. Thus, the lack of atmospheric feedback could explain extreme sublimation values on ridges (Sect. 4.2), contributing to their over-erosion. A partial coupling of FSM2trans with an atmospheric model such as HICAR (Reynolds et al., 2023) can be considered to further investigate this effect.

The most obvious limitation of SnowSlide is its non-dynamic representation of avalanche processes, which makes it challenging to model large deposit areas due to big avalanches, although the new hysteretic features we have introduced partly mitigate that issue. Figure D1 in the Supplementary Material shows the snow depth distribution in slopes steeper than 40°, in the LIDAR data, in FSM2ref and in FSM2trans. The average snow depth in steep slopes is improved (roughly divided by 2, matching the LIDAR average), but the variability is degraded: no more strong accumulations are possible in steep slopes, while the roughness of steep rocky faces sometimes allows snow to be retained (Sommer et al., 2015).

## 6   Conclusions

The modelling of snow redistribution induced by wind or gravity becomes necessary at hectometric and finer resolutions in order to better represent the resulting snowpack heterogeneity, which strongly influences the snow hydrological regimes in mountainous catchments. This study presents the new strategy developed in the Swiss operational snow hydrology modelling framework to address this issue, aiming at an intermediate complexity solution to best represent the processes while maintaining operationally viable computational times.

The present work offers a novel combination of approaches compared to existing models. It builds on the existing physics-based snowpack model FSM2oshd (Mott et al., 2023). A new density-dependent layering was included to represent more realistically the snowpack stratigraphy without resolving its microstructure, providing in particular a finer layering at the surface to determine erodible snow. These developments allowed for the inclusion of redistribution modules adapted to the new layering features. Wind-induced snow transport and sublimation were modelled by the snowdrift module SnowTran-3D (Liston et al., 2007). The avalanche module SnowSlide (Bernhardt and Schulz, 2010) was also included to represent gravity-induced snow transport, with the addition of simple hysteretic features to enable more realistic runout distances. Meteorological input fields were downscaled from a weather forecast model, using the dynamical downscaling model WindNinja (Forthofer et al., 2014) for wind fields. The simulations of the new FSM2trans model at 25 m, 50 m and 100 m resolutions were compared to four spatially distributed snow depth datasets acquired by airborne LIDAR surveys, in order to assess the added value of redistribution modelling for capturing catchment snowpack heterogeneity.

The FSM2trans snow depth maps showed a remarkable improvement in the representation of strong snow accumulations resulting from the interplay of snowdrift and avalanche processes, in terms of deposit positions and amounts. This improvement was quantified by an original structural similarity analysis of snow depth maps. The erosion and deposition areas were generally well captured in terms of aspect and slope. The main shortcomings were identified as an overestimation of ridgetop erosion and an underestimation of the extent of depositional areas. Saltation and suspension transport, as well as avalanches, were shown to

be major contributors to the mass budget on the most wind-exposed slopes and at high elevations. Snowdrift sublimation had a much smaller overall effect, except for a locally significant contribution to ridgetop erosion. The snow depth distribution plots confirmed a significant enhancement of the variability compared to the reference simulations, with the FSM2trans distribution curve consistently better matching the measured distribution curve from the peak of winter to the end of the melt season. As the snow depth distribution curve is a key control of snowmelt dynamics, this is a promising outcome to better represent snow hydrological processes at catchment scale. Further research should quantify the actual impact of redistribution on modelled catchment snowmelt runoff.

The most realistic snow distribution patterns were obtained at 25 m resolution, but redistribution at 50 m and 100 m resolutions also had a positive effect on the snow distribution, making our approach viable for operational applications over large extents which cannot afford resolutions as high as 25 m. These model developments have a limited computational impact and remain feasible within an operational framework. The possible practical application at nation-wide scale yet needs to be clarified. Finally, further enhancements of the representation of physical processes to mitigate current modelling limitations can still be achieved within the current intermediate-complexity framework. An atmospheric downscaling model like HICAR (Reynolds et al., 2023) could provide precipitation fields accounting for local terrain effects and wind fields accounting for lee-side recirculation, as well as a potential atmospheric feedback on snowdrift sublimation, at reasonable computational costs. Associated with the representation of forest snow processes (Mazzotti et al., 2020), such studies show that the representation of physical processes can be implemented in operational setups with significant benefits.

*Code availability.* FSM2trans code is available at https://doi.org/10.16904/envidat.509. This code builds on existing models: FSM2 (Essery, 2015), FSM2oshd (Mott et al., 2023), SnowTran-3D (Liston et al., 2007) and SnowSlide (Bernhardt and Schulz, 2010).

*Author contributions.* LQ designed the study and made the modelling developments of FSM2trans. PM, TJ and LQ set up the WindNinja model and provided downscaled input fields. LQ post-processed the LIDAR data with help from RM. LQ performed the simulations and analysed the results, with input from TJ and RM. BC, GM, LQ, RM and TJ contributed to the development of the OSHD modelling framework. LQ wrote the manuscript, with feedback from all authors.

*Competing interests.* The authors declare that they have no conflict of interest.

*Disclaimer.* Grammar and vocabulary of some manuscript sentences were corrected using DeepL (https://www.deepl.com).

*Acknowledgements.* The authors thank E. Brändle for first tests and assessments of the WindNinja simulations, as well as the two anonymous reviewers for their detailed comments, which helped to improve the paper. The OSHD model (FSM2oshd) development and implementation was largely funded by the Swiss Federal Office for the Environment (FOEN).

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

**Appendix A: Elevation profile of snow depth**

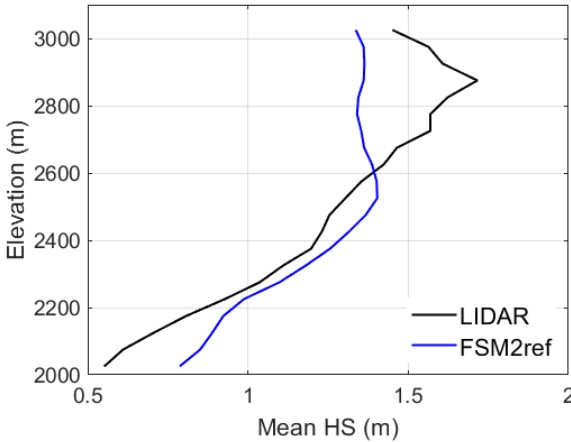

**Figure A1.** Elevation profile of mean snow depth on 20 March 2017 by 50 m elevation bands for subdomain D0, for LIDAR measurements (black) and FSM2ref simulations (blue).

 **Appendix B: Simulated snow depth maps at different spatial resolutions**

## B1  Maps at 50 m resolution

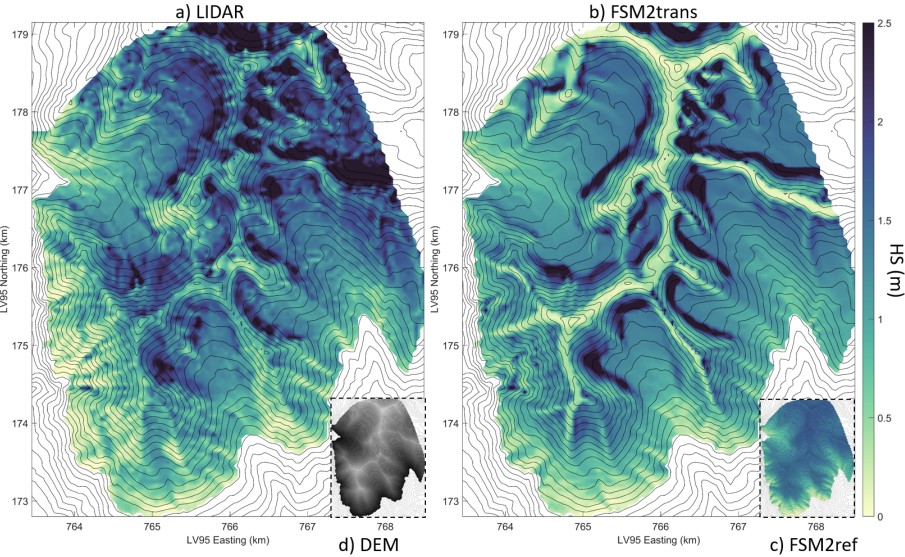

**Figure B1.** Map of snow depth on 17 March 2020 for subdomain B0: a) as measured by LIDAR and aggregated to 50 m resolution, b) as simulated at 50 m resolution by FSM2trans, c) as simulated at 50 m resolution by FSM2ref. d) Indicative DEM of subdomain B0 with higher elevations in lighter gray.

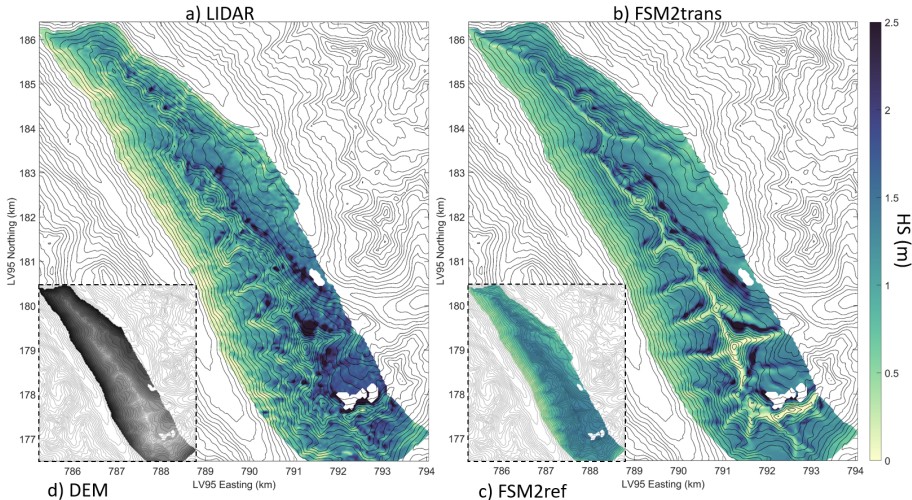

**Figure B2.** Map of snow depth on 31 March 2017 for subdomain D2: a) as measured by LIDAR and aggregated to 50 m resolution, b) as simulated at 50 m resolution by FSM2trans, c) as simulated at 50 m resolution by FSM2ref. d) Indicative DEM of subdomain D2 with higher elevations in lighter gray.

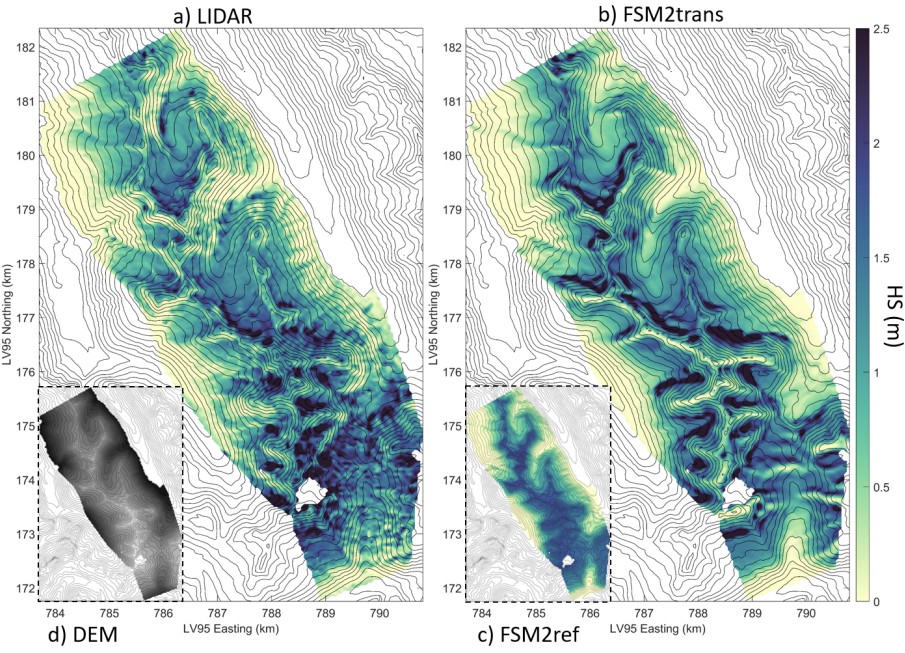

**Figure B3.** Map of snow depth on 17 May 2017 for subdomain D1: a) as measured by LIDAR and aggregated to 50 m resolution, b) as simulated at 50 m resolution by FSM2trans, c) as simulated at 50 m resolution by FSM2ref. d) Indicative DEM of subdomain D1 with higher elevations in lighter gray.

## B2   Maps at 100 m resolution

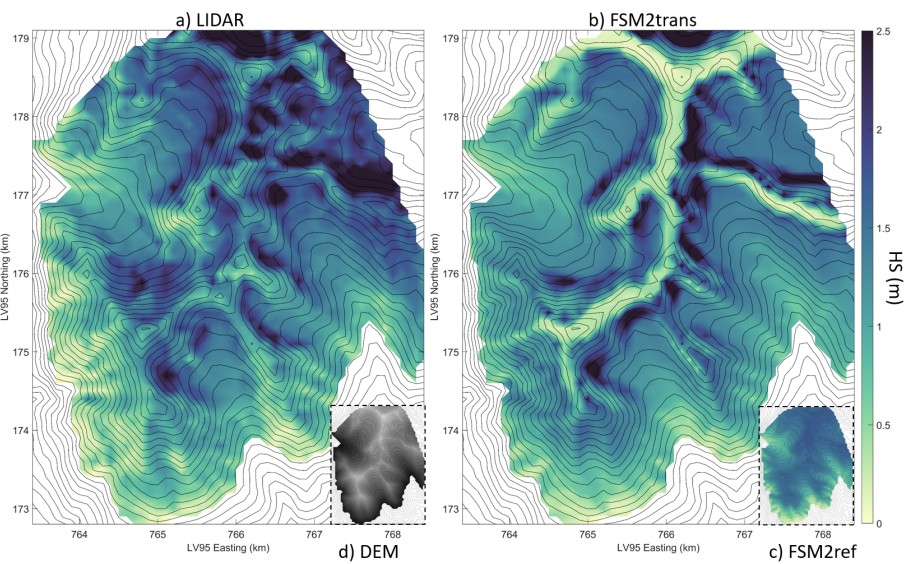

**Figure B4.** Map of snow depth on 17 March 2020 for subdomain B0: a) as measured by LIDAR and aggregated to 100 m resolution, b) as simulated at 100 m resolution by FSM2trans, c) as simulated at 100 m resolution by FSM2ref. d) Indicative DEM of subdomain B0 with higher elevations in lighter gray.

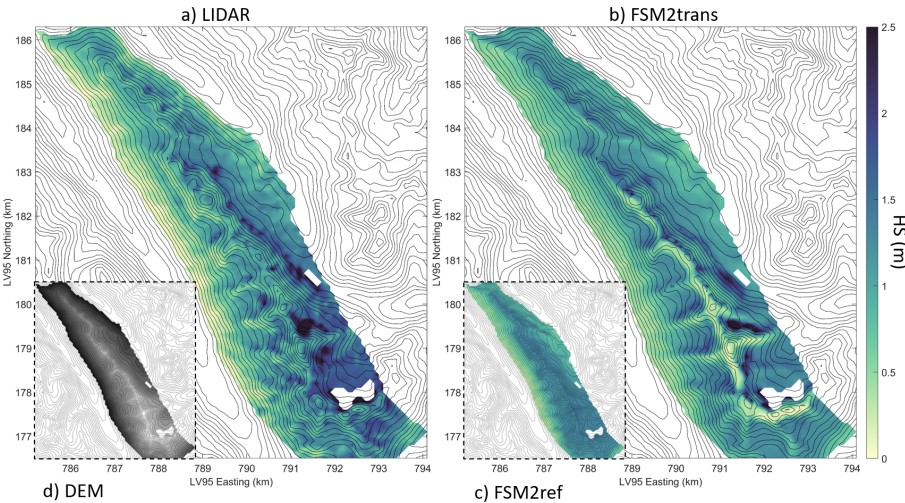

**Figure B5.** Map of snow depth on 31 March 2017 for subdomain D2: a) as measured by LIDAR and aggregated to 100 m resolution, b) as simulated at 100 m resolution by FSM2trans, c) as simulated at 100 m resolution by FSM2ref. d) Indicative DEM of subdomain D2 with higher elevations in lighter gray.

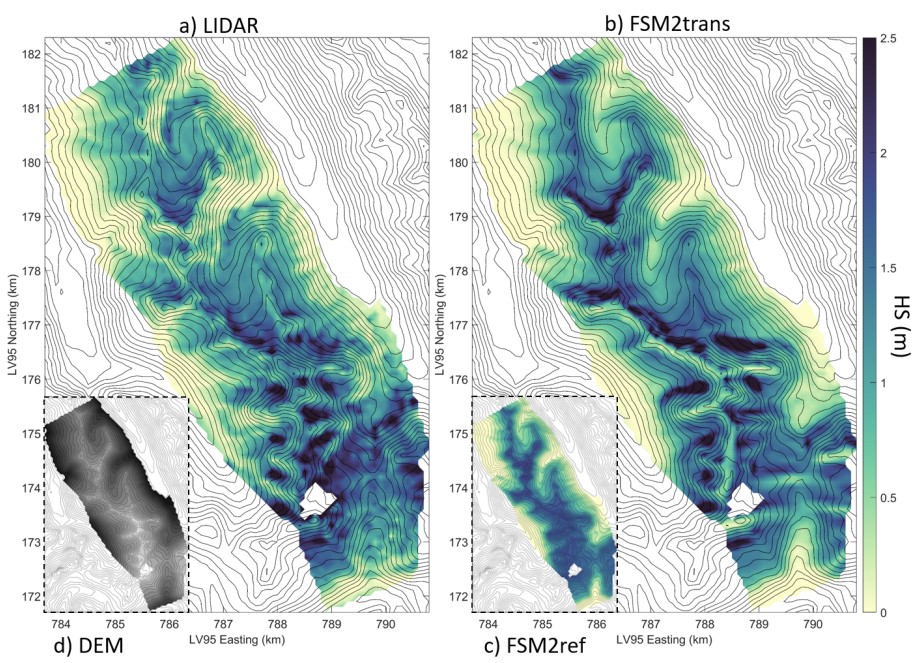

**Figure B6.** Map of snow depth on 17 May 2017 for subdomain D1: a) as measured by LIDAR and aggregated to 100 m resolution, b) as simulated at 100 m resolution by FSM2trans, c) as simulated at 100 m resolution by FSM2ref. d) Indicative DEM of subdomain D1 with higher elevations in lighter gray.

**Appendix C: Snow depth frequency plots at different spatial resolutions**

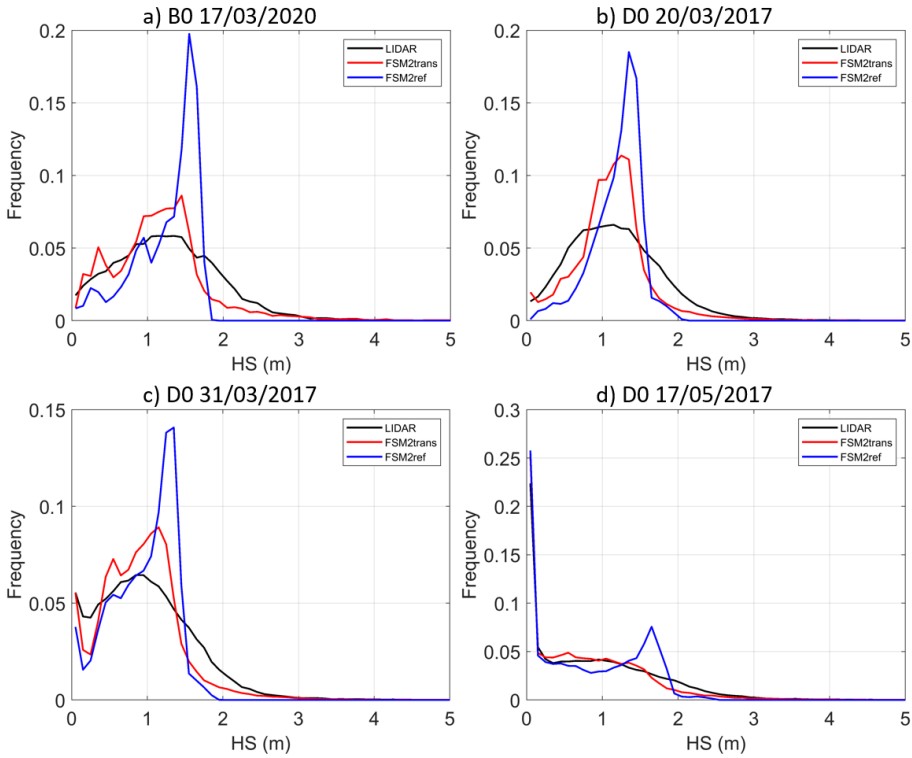

**Figure C1.** Snow depth frequency plot by intervals of 10 cm, at 50 m resolution, for LIDAR measurements (black), FSM2trans simulation (red) and reference FSM2ref simulation without redistribution (blue), for subdomain: a) B0 on 17 March 2020, b) D0 on 20 March 2017, c) D0 on 31 March 2017, d) D0 on 17 May 2017.

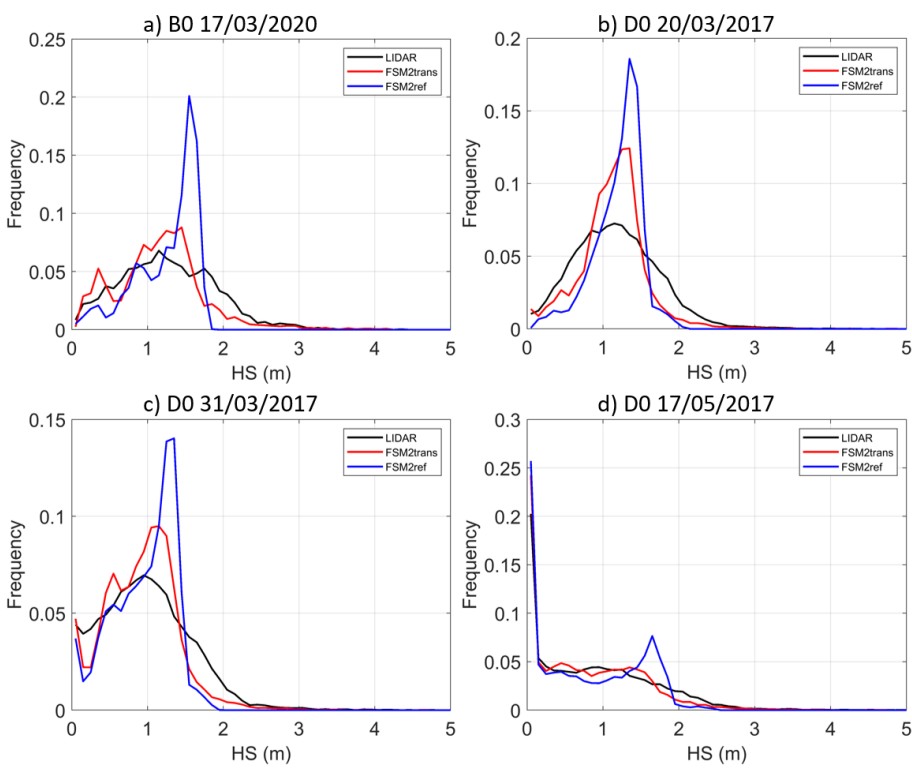

**Figure C2.** Snow depth frequency plot by intervals of 10 cm, at 100 m resolution, for LIDAR measurements (black), FSM2trans simulation (red) and reference FSM2ref simulation without redistribution (blue), for subdomain: a) B0 on 17 March 2020, b) D0 on 20 March 2017, c) D0 on 31 March 2017, d) D0 on 17 May 2017.

**Appendix D: Snow depth distribution on steep slopes**

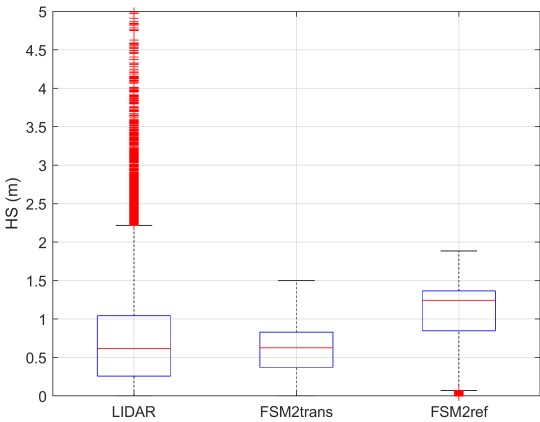

**Figure D1.** Boxplot of snow depth distribution on 31 March 2017 aggregated over slopes steeper than 40° for subdomain D0, for LIDAR measurements, FSM2ref and FSM2trans simulations.