# Peer review of "Snow redistribution in an intermediate-complexity snow hydrology modelling framework"

_EGUsphere, 2023_

## Author Comment (AC1)

**Answer to Referee #1**

We thank the referee for their insightful comments. Please find our detailed response to the issues raised by the reviewer below. Referee comments are in italics while our answers are in blue.

*General comments:*

*Congratulations to this very interesting paper! It brings the front of snow hydrological research one step further. The combination of the different modelling approaches is a valuable effort to combine methods, each of which as appropriate as possible for the scale, to ultimately integrate all relevant processes that determine the variability of snow depth in high mountain regions. Step by step we are coming closer to a snow hydrological model which allows robust prediction of snowmelt dynamics and, maybe even more important, of climate change effects on the snow distribution and its melting regime when combined with predictions from convection permitting climate models. This paper is an important contribution to this endeavour.*

*From my point of view, three issues desire some attention prior to finalizing the manuscript. The rest are minor comments.*

*The English is very good, I only found few details in the text where I suggest an alternative formulation.*

*Specific comments:*

*1) I recommend the authors to add a paragraph for the integration of the models and their timing: how were the submodels parameteriized (wind-induced snow redisribution, avalanches)? Does this parameterization depend on the scale (model/DEM resolution)? What triggers an event (blowing snow, avalanche)? What is the order of the computations in a time step, does this play a role? If yes, why is the chosen order the better one? These are all interesting questions for modellers and should be presented at least briefly.*

Following the referee's suggestion, a paragraph has been added in Section 2.3 to describe the sequence of processes of the redistribution modules, and technical choices that can be useful for modellers. More details have also been added in each module's description to explain more clearly chosen parameterizations.

*2) To my knowledge, SnowSlide updates the DEM surface elevation after each redistribution event with the accumulated mass of snow, thereby filling depressions and/or building up snow depositions in the runout zones of an avalanche. Isn't this the feature in SnowSlide that controls the runout area size of the snow if another (one after the other, actually) avalanche flows down the same slope/couloir (apart from parameters like maximum accumulation per pixel etc.)? This should be discussed in chapter 2.3.2., together with the new „hysteretic feature" (in a bit more detail).*

In its original form, SnowSlide updates the DEM with the updated snow depth at each time step, which enables to update the slope and the order of pixel calculations sorted by decreasing elevations. This version has been tested, which showed no significant visible differences in avalanche deposition areas. However, the calculation at each time step (i.e. every hour) of the new sorted elevation list takes a significant amount of time, superior to the modelling itself for large domains/highest resolutions

(complexity of order N log(N) to sort a list of length N). Consequently, with a view to intermediate complexity modelling applicable to operations, we have decided to discard this step. The implementation of the hysteretic features showed a more significant impact on the avalanche extents. These modelling choices have been more clearly explained in the revised manuscript.

*3) it would be nice to (make an attempt at least to) to evaluate the results of the single process simulations: solid precipitation amount, the new layering scheme (wetting events, density of the modelled snow layers), the modelled wind-induced lateral snow redistribution and the modelled avalanches as well.*

Following the reviewer's suggestion, additional FSM2trans simulations were performed with either only wind-driven redistribution, or only gravity-driven redistribution. In addition, following the other referee's suggestion, a more quantitative validation has been introduced to support statements in the comparison of modelled snow depth maps to the LIDAR datasets. The Structural Similarity Index (Wang et al., 2004) is used in the revised version of the manuscript to quantify the maps similarity combining similarities of luminance, contrast, and structure for each pixel with a chosen Gaussian radius (here set to 150 m). Snow depth maps can be compared as grey-scale images, where the snow depth is the luminance on a scale of 0 to 5 m. This metric has been computed for all simulations (a random snow depth distribution having a score of 0, an identical image having a score of 1), including wind transport only and avalanche only, which allows to evaluate the relative impact of each process simulation.

A figure showing the vertical profiles of snow depth (for LIDAR and for FSM2ref) has been added in the Supplementary Material to support the statement of lack of precipitation at highest elevations. References to results of Mott et al. (2023), showing similar precipitation trends, have been added.

*Technical corrections:*

*- 14-15: „… from the peak of winter to the end of the melt season": but not before peak of winter? Why? This should be mentioned here*

We refer here to our evaluation results which cover the period from peak of winter to end of melt season, which integrates redistribution processes throughout the whole winter. We choose to not add more details in the abstract to keep it concise.

*- Figure 1: the left panel should be larger (same size as the right one)*

Both maps have been edited to improve readability.

*- 95: „mostly in open terrain": what about forests, are these omitted here? There is probably a good reason for this, but it also should be expressed here*

Forests and urbanized areas were excluded from the study to focus on redistribution processes in open terrain. It has been clarified in that section.

*- 116, 142 and 151: I recommend to insert a table here with all existing FSM versions, including the original(s) by Richard Essery and all the follow-ups, including their names, references and main differences*

A table summarizing all model versions has been inserted in the revised manuscript.

*- 152-177: it would be nice for the reader if you show the effect of the two processes by means of an example simulation for a small but typical sub-area of one of your domains*

We agree with the reviewer that such local examples can allow the reader to understand more intuitively how the models work. However, in order to maintain the conciseness of the paper and to keep the number of figures to an acceptable level (after revision additions), we have decided not to add this supplementary figure, as the effect of both modules (separated by saltation/suspension, snowdrift sublimation and avalanche processes) is already illustrated by Fig. 9.

*- 169-177: are you using a SnowSlide version that updates the DEM surface elevation after each simulated transport event (i.e., adds deposited snow to a new surface elavation so that the next avalanche flows over it) to prevent „endless" increase of snow depth in depressions? See specific comment No. 2.*

Please see our answer to the second specific comment.

*- 160: are these „adaptions and improvements" that are discussed in the following? Maybe this could be made clear here*

Indeed, they are the adaptations and improvements described in the following sentences. It has been clarified.

*- 169: maybe better „using" instead of „offering"*

Corrected.

*- 170: is the „snow holding thickness" a snow depth threshold? The it should be mentioned here. A more general term would be „snow holding capacity".*

The snow holding capacity is defined as a threshold in snow thickness (i.e. normal to the slope), dependent on the slope. It has been clarified in the revised manuscript.

*- 171ff: how did you tune the SnowSlide parameters? See my specific comment No. 1.*

We used the parameterization of the snow holding capacity function of slope as implemented in the SnowSlide module of the Canadian Hydrological Model CHM (Marsh et al., 2020). It has been explicitly mentioned in the revised manuscript.

*- 174: are the „few improvements" the ones presented in the following?*

Indeed, they are the improvements described in the following sentences. It has been clarified.

*- 176: „extent": this means a larger deposition area, right? If yes, why not name it like this?*

The reviewer is right, it has been corrected.

*- 199-209: what can you say about the accuracy of the LIDAR-derived dataset? See my specific comment No. 3.*

The four LIDAR snow depth datasets were validated against manual snow depth measurements.

For domain D0 (20 March 2017, 31 March 2017 and 17 May 2017), a validation against more than 11 thousand manual measurements showed a bias of -4 to 0 cm and a RMSD (Root-Mean-Square Deviation) of 4 to 8 cm (Mazzotti et al., 2019).

For domain B0 (17/03/2020), a validation against 79 manual measurements showed a bias of - 2 cm and a RMSD of 15 cm.

This information about the accuracy of the LIDAR-derived dataset has been added in the revised manuscript.

*- 203: could you indicate explicitly earlier that you limit simulations to non-forested areas (see comment to line 95)?*

As suggested earlier, we clarified it in the "Modelling domain" section.

*- 204: 31 March 2017 is also covering the melting period?*

Indeed, melt had started at the lowest elevations of the domain on 31 March 2017. It has been clarified.

*- 209: evtl. better „aggregated to"*

Corrected.

*- 216: better „by" Winstral et al. (2017) and Dujardin and Lehning (2022)*

Corrected.

*- 220: you have both „snow depth" and „snowdepth" throughout the text. The former one is correct*

Corrected.

*- 225: probably better „for" subdomain B0 (all through the text where this occurs), and „while Fig. 4 shows subdomain" …*

Corrected.

*- 232: what do you mean with „spatialized" snow depth measurements, an interpolation result?*

The wording was indeed wrong. We meant "spatially distributed snow depth measurements", like the LIDAR snow depth dataset, as opposed to point snow depth measurements. It has been reworded in the revised manuscript.

*- 233: better „produces too little snow"*

Corrected.

*- 236: does „deposit extent" refer to area or mass, or both? I also think that it would better be „deposition" than „deposit"*

"Deposit extent" referred to the area. We have replaced it by "deposition area" for clarity.

*- 238: probably „accumulations" should better be singular, because it refers to the general nature of the process; or do you mean specific events?*

It has been corrected to "accumulation".

*- 241: here „accumulations" probably means „accumulated mass"?*

Indeed. It has been corrected.

*- 243: what are the „new hysteretic features of the avalanche model"? Maybe the slope threshold application mentioned in Sect. 2.3.2.? This deserves a more detailed explanation (see comment to lines 169-177 and specific comment No. 1)*

Section 2.3.2 has been clarified, following the reviewer's previous suggestion.

*- 246: I think it should be „spring" (lowercase; everywhere)*

Corrected.

*- 254: what do you mean with „resolutions … are irrelevant"? How can a resolution be irrelevant? Eventually you mean that the simulation results achieved for these resolutions do not properly reproduce redistribution processes …*

The wording was indeed unclear. We meant that going down to high resolutions such as 25 m, 50 m or 100 m does not bring significant added value compared to lower resolutions if redistribution processes are not modelled, given that the most significant part of variability at these scales is due to redistribution. We rephrased it in the revised manuscript: "simulations that do not include redistribution processes cannot represent a significant part of the snowpack spatial variability at 25 m resolution".

*- 258: is the reason for this the precipitation interpolation method the increase with altitude (the lapse rate)?*

The precipitation input is derived from interpolated 1 km resolution fields of the Numerical Weather Prediction model COSMO. Snowfall estimates are improved by data assimilation of snow depth measurements through optimal interpolation (Magnusson et al., 2014). However, these measurements are rare at high elevations (typically > 2500 m), which represent a significant part of our study domains. Underestimated precipitation at high elevations has already been noted earlier (Mott et al., 2023). It has been clarified in the revised manuscript.

*- 265: find something better than „over the whole subdomains" (what exactly do you mean with it, areas with TPI≤200?)*

The sentence has been reformulated: "The match of FSM2trans with the LIDAR is even better than when all TPIs are considered, with a clear improvement compared to FSM2ref.".

*- 272: what do you mean with „global", maybe „regional" or „in general"?*

It has been replaced by "the snow depth frequency curve", for clarity.

*- Figure 3: better „Map of snow depth on 17 March…", „for" subdomain … and aggregated „to". An image showing the difference between a) and b) would be very informative for the reader because it shows the spatial pattern…*

Corrected.

We deliberately decided not to show the snow height difference map. Indeed, a pixel-to-pixel bias can lead to a double penalty effect. For example, a correct snow transport extending one pixel further than the observation can generate a pixel of strong negative bias next to a pixel of strong positive bias, while the overall process is well represented. The resulting bias map is difficult to interpret and can be misleading on the actual model performance. This is the reason why we show distribution frequency plots and aggregations by topographic classes.

*- Figure 4: same as for the caption of Figure 3*

Corrected.

*- Figure 5: same as for the captions of Figures 3 and 4*

Corrected.

*- Figure 10: better „aggregated for the whole domain"*

Corrected.

*- chapter 5.2: see specific comment No. 3.*

The result and discussion sections of the revised manuscript have been enriched with a more quantitative analysis of map comparisons, with additional insights on the performance of wind transport and avalanches. Please see the answer to the third specific comment.

**References**

Magnusson, J., Gustafsson, D., Hüsler, F., and Jonas, T.: Assimilation of point SWE data into a distributed snow cover model comparing two contrasting methods, Water Resour. Res., 50, 7816–7835, https://doi.org/10.1002/2014WR015302, 2014.

Marsh, C. B., Pomeroy, J. W., and Wheater, H. S.: The Canadian Hydrological Model (CHM) v1.0: a multi-scale, multi-extent, variable complexity hydrological model – design and overview, Geosci. Model Dev., 13, 225–247, https://doi.org/10.5194/gmd-13-225-2020, 2020.

Mazzotti, G., Currier, W. R., Deems, J. S., Pflug, J. M., Lundquist, J. D., and Jonas, T.: Revisiting snow cover variability and canopy structure within forest stands: Insights from airborne lidar data, Water Resour. Res., 55, 6198–6216, https://doi.org/10.1029/2019WR024898, 2019.

Mott, R., Winstral, A., Cluzet, B., Helbig, N., Magnusson, J., Mazzotti, G., Quéno, L., Schirmer, M., Webster, C., and Jonas, T.: Operational snow-hydrological modeling for Switzerland, Front. Earth Sci., 11, https://doi.org/10.3389/feart.2023.1228158, 2023.

Wang, Z., Bovik, A. C., Sheikh, H. R., and Simoncelli, E. P.: Image quality assessment: from error visibility to structural similarity, IEEE Trans. Image Process., 13, 600–612, https://doi.org/10.1109/TIP.2003.819861, 2004.

---

## Author Comment (AC2)

**Answer to Referee #2**

We thank the referee for their insightful comments. Please find our detailed response to the issues raised by the reviewer below. Referee comments are in italics while our answers are in blue.

*This paper presents the development of a new modelling framework to simulate snow redistribution in mountainous terrain that can be applied over large domains with limited computational times. Such system is needed in the context of operational modelling of mountain snow hydrology in Switzerland. The authors give first an overview of the modelling system that accounts for wind-induced and gravitational snow transport. Its capacity to simulate snow distribution in mountains is then evaluated over a simulation domain surrounding Davos in Switzerland. Maps of snow depth derived from airborne LIDAR are used as a reference. The results show that model can generate realistic patterns of snow accumulation in mountainous terrain, including snow-free ridges, enhanced accumulation at the bottom of steep slopes, … Strong improvements are found in the distribution of snow depth close to peak snow accumulation and these improvements persist during the melting season. The main simulations of the paper were carried out at 25-m grid spacing. Additional simulations at 50- and 100-m grid spacing showed that improvements in snow distribution were also found at these resolutions, opening interesting opportunities for future operational system.*

*The subject of this paper is very relevant for the mountains snow hydrology community and the results shown here suggest that simulations including snow redistribution could be soon used in an operational context. The paper is well written, easy to follow and should ultimately be published in The Cryosphere. However, prior to publication, the authors should strengthen the results section to avoid statements that are not well supported by the figures and tables presented in the paper. This work would also benefit from a more quantitative approach relying on error metrics when comparing the different simulations and the observations. These two general comments are described first and are then followed by more specific and technical comments.*

*General comments*

*1. The results section of this paper starts with a comparison between simulated snow depth and observations from airborne Lidar (Section 4.1). This section is purely based on the visual comparison of maps (Fig 3 to 5) and probability distribution functions (PDF) of snow depth (Fig. 6 to 8). This section contains several statements that are not well supported by the results presented in these different figures. I recommend the authors to carefully revise this section and to remove the unsupported statements. Some of them can certainly be detailed introduced later in the text (in the discussion section for example), once more quantitative results have been presented (see my second general comment).*

Following the reviewer's suggestion, some of the unsupported statements were moved to the discussion section. We have also attempted to better justify the statements exposed by the reviewer below. Please see the detailed response to each of the four problematic statements.

*The first statement concerns the impact of combined snowdrift and avalanche modelling (P 9 L 230-232). I fully agree with this statement, but I find that it is not well supported by the results shown on*

*the two maps discussed here. It could have been better illustrated by considering simulations that consider only avalanching or wind-induced snow redistribution. I think Figure 9 helps to illustrate this interplay and the authors could make this statement later in the paper.*

Following the referee's suggestion, and in line with the other referee's suggestion, additional FSM2trans simulations were performed with either only wind-driven redistribution, or only gravity-driven redistribution. This allowed us to better assess the relative impact of each process and their combined impact. In particular, following the second general comment, an additional quantitative validation was introduced to strengthen the snow depth map comparisons and associated statements. This quantitative validation was applied to the two new simulation setups. Please see the response to the second general comment for more details on this newly added metric.

*A second statement is then made about the influence of the precipitation forcing (P9 L 233). At this stage of the analysis, it is not clear at all that the precipitation forcing can explain the underestimation of FSM2trans at the highest elevations. For example, Figure 3c does not suggest clearly that FSM2ref underestimates the snow depth at high elevations. A comparison of simulated and observed distribution of snow depth as a function of elevation could be used to show that FSM2ref (without redistribution) underestimates the snow depth at high elevation. This would strengthen the statement about the precipitation forcing. At this stage, it is not clear if this underestimation of snow depth is due to an overestimation of the intensity of wind-induced snow transport over exposed ridges in FSM2trans.*

We agree with the reviewer that this statement on the underestimation of precipitation at high elevations was not properly justified. As suggested, an elevation profile of snow depth for FSM2ref and the LIDAR dataset has been added to the Supplementary Material, showing a strong snow depth underestimation at elevations above 2600 m, despite the absence of simulation of redistribution and snowdrift sublimation. Moreover, references to Mott et al. (2023) were added to support this statement which was already identified in that paper. Snow depth measurement stations used for the data assimilation scheme are rare at high elevations, which can explain this bias. These elements supporting the statement have been added to the revised manuscript.

Finally, as pointed out by the reviewer, this precipitation underestimation is not the only cause of excessive wind-induced erosion on ridges. It has been clarified in the revised manuscript.

*A third statement explains that certain features of snow accumulation are due to" the new hysteretic features of the avalanche model" (P9 L 243). How would they look without the new features? These features are described in Section 2.3.2 but the motivations behind this development are never explained in the paper. A figure that shows patterns of avalanche deposition in the default and in the revised version of SnowSlide would be useful to understand why the revised version should be used in step alpine terrain. It could certainly be added in the supplementary material.*

The reviewer is right that the motivation for the development of new hysteretic features was not clearly explained. In its original form, SnowSlide updates the DEM with the updated snow depth at each time step, which enables to update the slope and the order of pixel calculations sorted by decreasing elevations. This version has been tested, which showed no significant visible differences in avalanche deposition areas. However, the calculation at each time step (i.e. every hour) of the new sorted elevation list takes a significant amount of time, superior to the modelling itself for large domains/highest resolutions (complexity of order N log(N) to sort a list of length N). Consequently, with a view to intermediate complexity modelling applicable to operations, we have decided to discard

this step. The implementation of the hysteretic features showed a more significant impact on the avalanche extents. These modelling choices have been more clearly explained in the revised manuscript.

Given these new detailed explanations about the introduction and parameterization of the hysteretic features in the revised manuscript, we have decided not to add another figure to compare both versions to keep the number of figures to an acceptable level and maintain the focus and conciseness of the paper.

*A fourth statement affirms that "FSM2ref can capture the average state of the snowpack over the subdomains" (P 9 L 253-254). and it is not clear at this stage of the paper. Quantitative metrics are required to show that that the average state of the snowpack is indeed well captured by FSM2ref (see my second general comment). In addition, L 254 refers to simulations at 50 and 100 m whereas no result from these simulations have been presented at this stage of the analysis.*

This statement indeed needed more justifications and nuances. References have been added to results and figures of Mott et al. (2023) who showed a good agreement of FSM2oshd with station measurements at all elevation bands. The statement was nuanced for very high elevations where: 1) very little validation data is available, 2) precipitation is likely underestimated (see previous response).

References to the simulations at 50 and 100 m resolutions at line 254 were deleted.

*2. Figure 6 to 8 show very convincing improvements in the ability of the model to simulate snow distribution in alpine terrain. However, at this stage, the comparison is purely qualitative. A more quantitative approach would significantly improve the paper. It could be used when (i) comparing FSM2ref and FSM2trans (P9 L 250-255), (ii) comparing the results for the full sub-domains and for ridges only (P11 L 265) and (iii) discussing the impact of the model grid spacing (P 9 L 254-255; P11 L 268-273). The visualization developed for Figure 10 could be used to present the distribution of error metrics (bias or RMSE for example) as a function of the elevation and orientation of the grid cells.*

Following the referee's comment, a more quantitative approach was added to better assess the comparisons of simulated snow depth maps with the LIDAR dataset. The Structural Similarity Index (Wang et al., 2004) is used in the revised version of the manuscript to quantify the maps similarity combining similarities of luminance, contrast, and structure for each pixel with a chosen Gaussian radius (here set to 150 m). Snow depth maps can be compared as grey-scale images, where the snow depth is the luminance on a scale of 0 to 5 m. This metric has been computed for all simulations (a random snow depth distribution having a score of 0, an identical image having a score of 1), including wind transport only and avalanche only, which allows to evaluate the relative impact of each process simulation.

*Specific Comments*

*P2 L 57: note that Liston et al. (2020) have developed a multi-layer version of SnowModel.*

This reference has been added to the revised manuscript.

*P 3 L 66-70: it would be interesting to mention here the recent developments of deep learning methods to downscale wind in complex terrain and to provide forcing to blowing snow scheme. See for example Le Toumelin et al. (2023).*

It is now mentioned, with the suggested reference.

*P3 L 75-76: Could you mention here feedback from users that have pointed out the limitations associated with the absence of snow redistribution in the operational model used at OSHD?*

A sentence has been added to mention the need for simulations representing slope scale variability, for users such as the avalanche warning service.

*P3 L 79: a distributed version of SnowModel has been recently applied at 100-m grid spacing over the contiguous Unites States by Mower et al (2023). The paper is still in discussion, but I still recommend the authors to add a sentence or two about this new implementation of SnowModel.*

This reference is now mentioned in the paragraph about recent studies.

*P4 L 97: what is the source of data used to generate the DEM at different resolutions?*

All DEMs used in this study were derived from the 25 m resolution digital height model (DHM25: https://www.swisstopo.admin.ch/en/height-model-dhm25, last access: 16/01/2024) of the Federal Office of Topography swisstopo. The data source has been included in the revised manuscript.

*P4 L 107: it would be interesting to add here a few sentences that describe how the OSHD version of FSM2 differs from the standard FSM2 version.*

The differences between FSM2 and FSM2oshd are extensively described by Mott et al. (2023). A sentence has been added to refer to this paper for more details on differences between the models.

*P 5 L 115: the authors have changed to layering in FSM2 to improve the simulation of surface snow properties and to better estimate snow erodibility. However, a change in the snow layering in a multi-layer snowpack model can also have an impact on the simulation of snow compaction, heat transfer and liquid water percolation through the snowpack, … Overall, can the authors comment on the impact of the new layering scheme on the simulation of seasonal snow evolution by FSM2? I guess it has been tested in the context of model development, especially if this version will ultimately replace the operational version of FSM2oshd.*

Snowpack simulations with the new layering scheme were compared to the operational simulations with fixed layering, showing very similar seasonal evolution, except for slightly increased settling and melting, most likely due to the presence of finer layers. Given that the operational model's parameters are tuned every year and that the seasonal dynamics were close, this difference was not judged significant. A sentence has been added in the paragraph to mention it.

*P5 L 116: the readers need to understand the novelty of the changes made to FSM2. For this reason, I recommend adding a short description of the original layering scheme used in FSM2. It will allow the reader to understand why such a scheme was not appropriate to represent the properties of surface and near-surface snow that are crucial when simulating snow transport.*

A sentence has been added in the paragraph to further explain the default layering of FSM2 used in FSM2oshd (fixed 3 layers, where the top two layers are 10 and 20 cm), and the associated limitations to simulated snow transport, in particular the representation of surface snow and the too thick layers to capture fine refrozen layers for example.

*P 5 L 131: a few sentences describing the regridding steps (conservation of mass, energy, …) would be useful.*

Relayering steps to maintain mass and energy conservation are now briefly mentioned after the description of the new layering scheme. These steps follow the same principles as the original FSM2 relayering scheme.

*P6 L 153: I am not familiar with the code management of SnowTran3D but, if possible, I recommend adding the version number of SnowTran3D that has been used when implementing it into FSM2trans.*

The code itself of SnowTran-3D was largely modified when adapted to the FSM2oshd framework. It follows the scientific principles and formulations exposed by Liston et al. (2007), except explicit mentions of modifications in our paper. The GitHub repository of FSM2trans will be made available in the published version of the manuscript.

*P 6 L 156-157: it would be interesting to add a few references describing the application of SnowTran3D at these resolutions.*

A few references have been added as suggested.

*P6 L 160-162: I am not sure to understand this sentence. Do the authors mean that the threshold friction velocity in the original SnowTran3D is computed using a constant density? Consider rephrasing this sentence.*

The sentence was indeed unclear. Our model performs snow erosion one layer after the other from the top of the snowpack, where the eroded snow depth is calculated based on each layer's density. Originally, the eroded/accumulated depth was calculated based on the net mass flux assuming a constant density. It has been clarified.

*P6 L 163-164: The default version of SnowModel described in Liston et al. (2007) includes a parameterization (Eq 18 in Liston et al., 2007) to simulate the increase of near-surface density due to fragmentation during blowing snow events. The influence of wind speed on near-surface density is also included in SnowModel through a wind-related density offset for fresh snow falling in windy conditions (Eq 16 in Liston et al., 2007). Is FSM2trans including these effects? If not, it should be explained clearly in the text. The absence of snow microstructure mentioned at L163 is not reason to justify the absence of compaction during snowdrift in FSM2trans.*

In FSM2trans, the top layer of the snowpack is indeed compacted under the influence of the wind, following equations 17 and 18 of Liston et al. (2007). However, the density of *redeposited* snow is set constant, as in SnowTran-3D. The sentence has been clarified in the revised manuscript.

*P6 L 170: Is the snow holding capacity considered in FSM2trans applied to the snow depth (measured vertically) or the snow thickness (measured perpendicular to the slope)?  Are the authors using the default formulation from Berhnard and Schulz (2010) for the holding depth?*

The snow holding capacity is applied to the snow thickness. It is then converted into a snow depth threshold through a cosine factor. We used the parameterization of the snow holding capacity function of slope as implemented in the SnowSlide module of the Canadian Hydrological Model CHM (Marsh et al., 2020). It has been clarified in the revised manuscript.

*P 7 L 187: Was a cosine correction applied to adjust precipitation based on the local slope of the grid cell for mass-conservation purposes (Kienzle, 2011)?*

We don't apply a cosine correction to adjust precipitation because the input precipitation is per pixel (vertical precipitation) and not per surface area. The model outputs the snow depth (vertical distance from base to snow surface) and not the snow thickness (normal to the slope). For a given precipitation amount per pixel, the snow depth is the same on a flat pixel and on a sloped pixel, while the snow thickness is lower on the sloped pixel. However, the cosine correction is applied to calculate the snow holding capacity defined as a threshold on snow thickness.

*P 7 L 190: Which formulation is used to split between rain and snow?*

The splitting between rain and snow follows the formulation used operationally in FSM2oshd (Mott et al., 2023), a sigmoid function based on the 10 m air temperature $Ta_{10m}$ (in °C):

$$\frac{snowfall}{precip_{tot}} = \frac{1}{1 + \exp\left(\frac{Ta_{10m} - 1.04}{0.15}\right)}$$

[Figure]

It is now mentioned in the revised manuscript.

*P 8 L 194: Was the wind downscaling done at model runtime? Or did the authors prepare downscaled wind fields for the whole season that were then used to drive FSM2trans and FSM2ref? It would be interesting to add a few sentences about the numerical cost of the wind downscaling since the main objective of this paper is to present a system that can be used in an operational context. The wind downscaling is a crucial step for the success of any modelling of snow redistribution in complex terrain.*

Downscaled wind fields were prepared separately for the whole season using WindNinja. The numerical cost of wind fields preparation has been added in the discussion together with FSM2trans runtimes.

However, note that the focus of the present study is the redistribution model, which is not constrained by the use of any specific wind downscaling method. We fully agree with the reviewer that it is a crucial step for snow redistribution modelling. A distinct paper will follow, focusing on the comparison of different wind downscaling methods in this framework, with an assessment of the different wind fields, their impact on redistribution modelling and their suitability for intermediate-complexity modelling in an operational context.

*P 8 L 209: How are treated the data that were masked out (glaciers, lakes, outliers) when computing the averaged snow depth at different resolution?*

Since the outliers were mostly outside of the area of interest (elevations > 2000 m) and very isolated above 2000 m, masked data at high resolution were simply excluded from the mean over aggregated pixels. Glacier masks were applied in a second step on top of the averaged snow depth maps.

*P 12 L 278: to better understand the maps shown of Figure 9 it would be interesting to have one or two sentences describing the dominant direction of the main blowing snow events in the region.*

The dominant wind directions (from NW to SW) were indicated in the revised manuscript to facilitate map interpretation.

*P 16 L 337: It would be interesting to add information about the numerical cost of the generation of the wind fields at different resolutions. Marsh et al. (2023) (Section 4.4) have shown that the stand-alone version of WindNinja can have a large numerical cost compared to a method based on pre-computed wind library.*

Following the reviewer's suggestion, the numerical cost of wind fields preparation has been added in the discussion together with FSM2trans runtimes. Please also refer to our previous response about wind downscaling.

*P 17 L 350-353: A figure illustrating the evaluation of wind speeds downscaled by WindNinja would be useful for the readers since the wind forcing is crucial when talking about wind-induced snow redistribution in complex terrain. What is the quality of the simulations for strong wind events that are driving wind-induced snow redistribution? I believe that in the context of this work a bias computed over a full month is less relevant than statistics about strong wind events.*

The preliminary evaluation of WindNinja is only presented here as an indicative element for discussing redistribution results. We fully agree that an extensive evaluation of WindNinja in mountainous terrain would be very useful to interpret results. However, as mentioned in a previous response, the evaluation of wind fields resulting from downscaling methods is not the focus of the present paper, addressing the redistribution model and its evaluation. A distinct paper will follow, focusing on the comparison of different wind downscaling methods in this framework, with an assessment of the different wind fields, their impact on redistribution modelling and their suitability for intermediate-complexity modelling in an operational context.

*P 17 L 360-365: Mott and Lehning (2010) found a similar overestimation of snow redistribution for a crest of the Swiss Alps using the Alpine 3D model running at 25 and 50 m grid spacing. They showed that increasing the model resolution finer than 10 m increased snow accumulation on the windward side due a more accurate representation of small-scale terrain features trapping snow on the windward side. Therefore, I am not sure that the lack of snow on ridges is only explained by a bias in the precipitation forcing. It can also be associated with limitations in the snow redistribution module.*

We fully agree with the reviewer: the precipitation forcing bias is not the only cause for the lack of snow on ridges. Scaling issues in the redistribution module are indeed a well-known and significant problem for representing snowpack variability. For example, while in reality ridges can be locally very eroded, the extent of this strong erosion is often on the order of 1 to 10 m across the ridge, whereas a redistribution model (without subgrid parameterization) will lead to strong erosion at the ridgetop pixel, which can extend to 25, 50 or 100 m depending on the model resolution, hence the discrepancies with observations. These limitations have been clarified in the revised manuscript.

*P 18 L 383: on this figure, are the authors comparing snow depth (measured vertically) or snow thickness (measured perpendicular to the slope)?*

The whole text is written in conformity with the international classification for seasonal snow on the ground (Fierz et al., 2009), i.e. snow depth (HS) represents the vertical distance from base to snow surface. This is the case for Fig. C1. The only occurrence of snow thickness (normal to the slope) is to define the snow holding capacity in the SnowSlide module. This has been clarified in the SnowSlide description section in the revised manuscript.

*Technical Comments*

*P1 L5: maybe add "the models" or "the module" before "SnowTran-3D and SnowSlide"*

Corrected.

*P1 L8: Use superscript for km2*

Corrected.

*P4 L100: Paragraphs made of one sentence should be avoided.*

Corrected.

*P 5 L123: New snow that accumulates from avalanches cannot be considered as fresh snow. Please rephrase the sentence.*

Corrected.

*Figures*

*Figure 1: The contours of Switzerland are hard to see on the first map. The contour of D2 in light green are also hard to read on the main map.*

Both maps have been edited to improve readability.

*Tables*

*References (used in this review and not present in the initial manuscript)*

*Kienzle, S. W.: Effects of area under-estimations of sloped mountain terrain on simulated hydrological behaviour: a case study using the ACRU model, Hydrol. Process., 25, 1212–1227, https://doi.org/10.1002/hyp.7886, 2011.*

*Le Toumelin, L., Gouttevin, I., Helbig, N., Galiez, C., Roux, M., & Karbou, F. (2023). Emulating the Adaptation of Wind Fields to Complex Terrain with Deep Learning. Artificial Intelligence for the Earth Systems, 2(1), e220034.*

*Liston, G. E., Itkin, P., Stroeve, J., Tschudi, M., Stewart, J. S., Pedersen, S. H., ... & Elder, K. (2020). A Lagrangian snow-evolution system for sea-ice applications (SnowModel-LG): Part I—Model description. Journal of Geophysical Research: Oceans, 125(10), e2019JC015913.*

*Mott, R. and Lehning, M.: Meteorological modeling of very high-resolution wind fields and snow deposition for mountains, J. Hydrometeorol., 11, 934–949, https://doi.org/10.1175/2010JHM1216.1, 2010.*

*Mower, R., Gutmann, E. D., Lundquist, J., Liston, G. E., and Rasmussen, S.: Parallel SnowModel (v1.0): a parallel implementation of a Distributed Snow-Evolution Modeling System (SnowModel), EGUsphere [preprint], https://doi.org/10.5194/egusphere-2023-1612, 2023*

**References**

Liston, G. E., Haehnel, R. B., Sturm, M., Hiemstra, C. A., Berezovskaya, S., and Tabler, R. D.: Simulating complex snow distributions in windy environments using SnowTran-3D, J. Glaciol., 53, 241–256, https://doi.org/10.3189/172756507782202865, 2007.

Marsh, C. B., Pomeroy, J. W., and Wheater, H. S.: The Canadian Hydrological Model (CHM) v1.0: a multi-scale, multi-extent, variable complexity hydrological model – design and overview, Geosci. Model Dev., 13, 225–247, https://doi.org/10.5194/gmd-13-225-2020, 2020.

Mott, R., Winstral, A., Cluzet, B., Helbig, N., Magnusson, J., Mazzotti, G., Quéno, L., Schirmer, M., Webster, C., and Jonas, T.: Operational snow-hydrological modeling for Switzerland, Front. Earth Sci., 11, https://doi.org/10.3389/feart.2023.1228158, 2023.

Wang, Z., Bovik, A. C., Sheikh, H. R., and Simoncelli, E. P.: Image quality assessment: from error visibility to structural similarity, IEEE Trans. Image Process., 13, 600–612, https://doi.org/10.1109/TIP.2003.819861, 2004.

---

## Author Response (AR2)

**Answer to Referee #1**

We thank the referee for their support and final remarks. Please find our response to their recommendations below. Referee comments are in italics while our answers are in blue. Corresponding changes in the revised manuscript are referenced using the lines and pages of the track-changes manuscript.

*Dear authors, well done! I accept almost all changes you did and the responses to my review. There remain only two very small recommendations for improvement:*

*- Figure 1: the left panel should be larger (same size as the right one)*

*„Both maps have been edited to improve readability.“*

*Sorry, but this is not much better. Wouldn't it be the best solution to place the two panels not vertically, but horizontally aligned, both of the same height, using the available space in the width? If you don't have the Swiss map picture in a printable resolution to further enlarge it you should search for an alternative. There should be plenty available.*

We increased the size of the Swiss map to the same width as the domain map. Placing the two maps side by side resulted in both maps being smaller, even when using the maximum width available. We hope this makes the map easier to read.

➢ CHANGES IN THE MANUSCRIPT: Figure 1 (P. 5)

*- Table 1*

*In the new table 1, you could include that FSM2 includes snow-canopy interaction process parameterizations (other than the original FSM).*

To avoid creating overload and confusion in the table, we have decided not to mention snow-canopy interaction processes there, but directly in the text of the revised version.

➢ CHANGES IN THE MANUSCRIPT: L. 113-114 (P. 4)

**Answer to Referee #2**

We thank the referee for their support and final remarks. Please find our response to their recommendations below. Referee comments are in italics while our answers are in blue. Corresponding changes in the revised manuscript are referenced using the lines and pages of the track-changes manuscript.

*In the revised paper, the authors have addressed very well the questions that I raised during the first stage of the review. The similarity metric that they have introduced allows a quantitative evaluation of the different model experiments and illustrate well the relative importance of wind-induced and gravitational snow transport and their interplay. This paper is an excellent contribution to the literature, and it should be published in TC. I have added below a few minor comments.*

*Line comments (with line numbers referring to the new version of the paper, in Track Change mode):*

*P 9 L 237-238: I recommend the authors to explicitly mention if WindNinja was used to simulate wind speed and direction at the three horizontal resolutions mentioned in the paper (25, 50 and 100 m) or if it has been only used to simulate wind fields at 25 m and that the wind fields at the other resolution have been obtained by averaging.*

For the present study, WindNinja was run separately at the three horizontal resolutions. We clarified it in the revised manuscript.

  ➢  CHANGES IN THE MANUSCRIPT: L. 235-236 (P. 9)

*P 10 L 275: did the author used a specific package (in Python, R, …) to compute the Structural Similarity Index? It could be interesting to mention it since I this metric will certainly be used by other researcher working in this field.*

To compute the Structural Similarity Index, we used the ssim function from the Matlab Image Processing Toolbox (https://www.mathworks.com/help/images/ref/ssim.html, last access: 8 May 2024). This metric has been implemented in several other languages, for example in Python using the scikit-image library (https://scikit-image.org/docs/stable/auto_examples/transform/plot_ssim.html, last access: 8 May 2024). We mentioned it in the revised manuscript.

  ➢  CHANGES IN THE MANUSCRIPT: L. 277-280 (P. 11)